# Convex Potential Mirror Langevin Algorithm for Efficient Sampling of Energy-Based Models

## Abstract

This paper introduces the Convex Potential Mirror Langevin Algorithm (CPMLA), a novel method designed to optimize sampling efficiency within Energy-Based Models (EBMs). CPMLA employs mirror Langevin dynamics in conjunction with convex potential flow as a dynamic mirror map for sampling in EBMs. By leveraging this dynamic mirror map, CPMLA enables targeted geometric exploration on the data manifold, enhancing the convergence process towards the target distribution. Theoretical analysis proves that CPMLA achieves exponential convergence with vanishing bias under relaxed log-concave conditions, supporting its efficiency and effectiveness in adapting to complex data distributions. Experimental results on established benchmarks like CIFAR-10, SVHN, and CelebA showcase CPMLA's enhanced sampling quality and inference efficiency compared to existing techniques.

## 1 Introduction

Energy-based generative models (EBMs) constitute a category of machine learning models employed for data generation. In contrast to traditional generative models that directly model data distributions, EBMs use an energy learning strategy that assigns low values to favorable samples and high values to unfavorable ones (Xie et al., 2016; Nijkamp et al., 2019; Du & Mordatch, 2019). Known for their simplicity and training stability, EBMs have found diverse applications ranging from 3D object recognition (Gustafsson et al., 2021) and analysis (Xie et al., 2018) to image segmentation (Kashyap & Gautam, 2017), super-resolution restoration (Zheng et al., 2021), machine translation (Tu et al., 2020), and protein folding (Tubiana et al., 2018; Wu et al., 2021).

EBMs encounter sampling challenges due to their dependence on computationally demanding Markov Chain Monte Carlo (MCMC) techniques in high-dimensional spaces (Barbu et al., 2020; Du & Mordatch, 2019; Kumar et al., 2019). The primary issue is the slow convergence that stems from the non-mixing phenomenon, i.e. the tendency to get trapped in local modes. These limitations make it difficult for MCMC-based approaches to efficiently approximate complex target distributions, especially when the data manifold is highly non-Euclidean (Zhang et al., 2020; Jiang, 2021).

Several recent methodologies have surfaced to address sampling inefficiencies within EBMs. While some strategies concentrate on refining MCMC initialization (Hinton, 2002; Du & Mordatch, 2019), others explore gradient approximation techniques (Kim & Bengio, 2016; Kumar et al., 2019). Despite these advancements, persistent challenges such as non-mixing issues remain unresolved (Xie et al., 2022b). Mirror Langevin algorithms have recently emerged as an alternative approach to alter sampling geometry via a fixed mirror map, i.e. a predefined function. Prior work (Ahn & Chewi, 2021; Li et al., 2022) demonstrates that mirror Langevin algorithms, under certain assumptions, exhibit vanishing bias, i.e. the bias approaches zero as the step size $h \to 0$. This property ensures that the samples reliably converge to the target distribution and improves sampling accuracy. Moreover, mirror Langevin algorithms produce samples with a mixing time that can be rendered independent of the condition number of the domain, enabling fast convergence (Hsieh et al., 2018; Zhang et al., 2020). However, the limitation of the fixed mirror map in efficiently capturing the data manifold has inhibited the exploration of this alternative sampling method for addressing large-scale problems especially those associated with deep neural networks.

This paper introduces Convex Potential Mirror Langevin Algorithm (CPMLA), a novel approach for sampling EBMs with enhanced efficiency. In contrast to conventional mirror Langevin algorithms that function within Euclidean space, CPMLA facilitates an efficient and effective sampling process via a dynamic mirror map, a parameterized function being optimized during the training process. Inspired by Brenier's theorem (Santambrogio, 2015), the dynamic mirror map is modeled via a parameterized convex potential function, whose gradient w.r.t. the input corresponds to the unique solution for the Monge problem between noise distribution and data distribution. While traditional mirror Langevin algorithms are commonly utilized for constrained sampling tasks with fixed mirror maps, CPMLA utilizes a dynamic mirror map to capture the intrinsic geometric complexities of the data manifold. This capability allows for adaptive adjustments in the sampling direction based on the data distribution, thereby augmenting the algorithm's overall effectiveness.

We employ a cooperative learning strategy to optimize our model. Initially, CPMLA learns the dynamic mirror map via optimizing a convex potential flow (CP-Flow) (Huang et al., 2020). Supported by Brenier's theorem (Santambrogio, 2015), CP-Flow, derived from the gradient of a convex energy function (Amos et al., 2017), can be optimized to capture the underlying data distribution. Subsequently, the EBM is trained by optimizing the energy difference between real data samples and those synthesized via CPMLA. Then, the synthesized samples by EBM are used to augment the training of the dynamic mirror map, i.e. CP-Flow. This iterative process fosters ongoing refinement and enhancement of the dynamic mirror map and the EBM.

We present a theoretical analysis of the convergence properties of the proposed CPMLA. Following the recent study (Jiang, 2021), we theoretically demonstrate our method can achieve a fast convergence ($\tilde{\Omega}\left(M\gamma^2 d/\beta^2\delta\right)$ mixing time) under relaxed log-concavity assumptions. In particular, we specialize our proof for the dynamic mirror map modeled with deep neural networks, which broadens its applicability to a wider range of target distributions in various machine learning tasks. Additionally, we streamline the proof by taking advantage of the bounded gradients property exhibited by the mirror map. To the best of our knowledge, this work is the first to analyze mirror Langevin algorithms within the framework of deep neural networks, resulting in exponential convergence with vanishing bias (Theorem 5.5).

We conducted a thorough evaluation of CPMLA across several standard benchmark datasets, including CIFAR-10, SVHN, and CelebA. The results demonstrate that CPMLA not only achieves superior sampling quality but also exhibits enhanced inference efficiency compared to existing cooperative algorithms. Specifically, CPMLA attained a FID score that is 27% of Flow+EBM's (Gao et al., 2019; Nijkamp et al., 2020), indicating a substantial improvement in visual quality. Additionally, CPMLA not only can achieve a FID score comparable to CoopFlow (Xie et al., 2022b) with fewer inference iterations, but also operates with only 0.9% of the parameter count, underscoring its remarkable efficiency in both sampling and inference. CPMLA has also demonstrated outstanding performance in specialized tasks such as image reconstruction and inpainting, further emphasizing its effectiveness in tackling complex image processing challenges.

Our main contributions are summarized as follows:

- We propose a novel Convex Potential Mirror Langevin Algorithm (CPMLA) for efficient sampling of EBMs. The efficiency comes from the modification of the sampling geometry through a dynamic mirror map modeled with a deep neural network.
- We provide a theoretical analysis of the convergence of the proposed CPMLA, focusing specifically on deep neural networks under relaxed assumptions.
- We evaluate the efficacy of our proposed algorithm through comprehensive experimental analyses on various benchmark datasets, including CIFAR-10, SVHN, and CelebA. Our experiments demonstrate that our CPMLA achieves superior sampling efficiency compared to existing methods. Furthermore, it surpasses alternative approaches in terms of sample quality and the fidelity of image reconstruction and inpainting.

## 2 RELATED WORK

**Langevin sampling** Numerous discretizations of Langevin dynamics within Euclidean geometry have been extensively studied in the literature, with non-asymptotic error bounds derived for various sampling error metrics, such as Kullback-Leibler divergence, Total Variation, and Wasserstein distance. The most extensively studied scenarios include cases where the target distribution is $m$-

strongly log-concave (Durmus & Moulines, 2016a;b; Cheng et al., 2017; Dalalyan & Karagulyan, 2019; Mou et al., 2019) and those where it is relaxed log-concave (Wibisono, 2019; Ma et al., 2021; Mou et al., 2022). Recently, mirror Langevin dynamics has garnered attention in the field of non-Euclidean geometry sampling, due to its superior convergence properties in constrained optimization problems. Originally introduced by Hsieh et al. (2018) as a measure transformation of the classical Langevin dynamics, the convergence of mirror Langevin dynamics under relaxed log-concavity was soon investigated in Zhang et al. (2020), where the authors demonstrated convergence to a Wasserstein ball with non-vanishing bias. In Ahn & Chewi (2021), under similar relaxed log-concave conditions, it was shown that the bias decreases to zero as the step size diminishes. Closely related to our work is Chewi et al. (2020), where the authors studied the convergence of the discretized process using functional inequalities similar to those we examine, though the practical applications were left for future exploration.

**Cooperative learning** The cooperative learning concept, first introduced in Xie et al. (2022a), involves the joint maximum likelihood training of a ConvNet-EBM (Xie et al., 2016) and a top-down generator (Han et al., 2022). In a similar vein, Xie et al. (2021) replaced the generator in the original CoopNets with a variational autoencoder (VAE) (Kingma & Welling, 2013) to improve inference efficiency. Our learning algorithm draws inspiration from the recent Coopflow approach (Xie et al., 2022b), which focuses on the collaborative training of a Langevin flow and a normalizing flow to elevate the quality of initial samples used in Langevin samplers. In contrast, our CPMLA excels in two key aspects: firstly, our sampling strategy delves into the data manifold by leveraging a convex potential; secondly, underpinned by mathematical principles, our CPMLA demonstrates faster convergence in scenarios of relaxed log-concave sampling, especially tailored for deep neural networks.

## 3 BACKGROUND

In this section, we elucidate the cooperative learning framework by introducing two interconnected components: EBM and convex potential flow. The convex potential flow is the dynamic mirror map of our CPMLA which is designed for efficient sampling of the EBM.

### 3.1 ENERGY-BASED MODELS

Let $x \in \mathbb{R}^D$ denote a sample. An energy-based model defines the probability density of $x$ as an exponential of the negative energy function, which is

$$p_\theta(x) = \frac{1}{Z(\theta)} \exp[f_\theta(x)] \tag{1}$$

where $f_\theta : \mathbb{R}^D \to \mathbb{R}$ is the negative energy function which is defined by a neural network parameterized by $\theta$, $Z(\theta) = \int \exp[f_\theta(x)]dx$ is a normalizing constant.

To generate synthesized examples from the distribution $p_\theta(x)$, it commonly uses the discretizations of Langevin dynamics, i.e. Langevin Monte Carlo (LMC) (Welling & Teh, 2011), which can be formulated as follows:

$$\hat{x}^{t+1} = \hat{x}^t + \frac{\delta^2}{2}\nabla_x f_\theta(\hat{x}^t) + \delta\varepsilon^t \tag{2}$$

where $\varepsilon^t \sim \mathcal{N}(0, I)$, $t$ is the Langevin time step, $\delta$ represents the stepsize of the LMC, and the initialization of LMC $\hat{x}^0$ is drawn from a uniform distribution $p_0(x)$.

Given a set of unlabeled training samples $x_i$, $i = 1, \cdots, n$ drawn from an unknown data distribution $p_{\text{data}}(x)$, we can update the model parameter $\theta$ by approximating the gradient of the log-likelihood function of the model $p_\theta(x)$ as follows:

$$\mathcal{L}'(\theta) = \mathbb{E}_{p_{\text{data}}}\left[\nabla_\theta f_\theta(x)\right] - \mathbb{E}_{p_\theta}\left[\nabla_\theta f_\theta(x)\right] \approx \frac{1}{n}\sum_{i=1}^n \nabla_\theta f_\theta(x_i) - \frac{1}{n}\sum_{i=1}^n \nabla_\theta f_\theta(\hat{x}_i) \tag{3}$$

where $\hat{x}_i, i = 1, 2, \cdots, n$ are synthesized examples drawn from the distribution $p_\theta(x)$ via LMC sampling. It is worth noting that the analytically intractable normalizing constant $Z(\theta)$ is approximated

by the average gradient of the energies from synthesized example $\hat{x}_i$. Subsequently, the learning rule above is based on the difference estimation between the energy gradient over the observed data distribution and the negative energy gradient over the synthesized examples.

## 3.2 Convex Potential Flow

Convex Potential Flow (CP-Flow) (Huang et al., 2020) is a flow-based model that aims to learn tractable probabilistic density. CP-Flow corresponds to an approximation of the gradient of the unique optimal transport for the classical Monge problem between the noise and target data distribution, as shown below.

**Optimal Transport** The Monge problem (Villani et al., 2009) revolves around the quest for an optimal transport map $g$ that achieves the minimum expected cost as follows:

$$J_c(p_X, p_Y) = \inf_{g:g(x) \sim p_Y} \mathbb{E}_{X \sim p_X}[c(x, g(x))] \tag{4}$$

where $c(x, y)$ is the given cost function.

**Theorem 3.1.** *(**Brenier's Theorem** (Santambrogio, 2015)) Suppose $\mu$ and $\nu$ are probability measures with finite second moments, and assume that $\mu$ has a Lebesgue density $p_X$. In this case, there exists a convex potential $G$ such that the gradient map $g = \nabla G$ (uniquely defined except for a null set) provides the solution to the Monge problem in Equation 4 with square cost function $c(x, y) = ||x - y||^2$.*

To approximate the optimal solution for the Monge problem, the convex potential is modeled with several layers of input-convex neural network (ICNN) $G_\vartheta$ (Amos et al., 2017), which is convex w.r.t the input:

$$\begin{aligned} G_\vartheta(x) &= L_{K+1}^+ (s(z_K)) + L_{K+1}(x) \\ z_k &:= L_k^+ (s(z_{k-1})) + L_k(x), \quad z_1 := L_1(x) \end{aligned} \tag{5}$$

where $\vartheta$ denotes the parameters of $G$, $L(x)$ denotes a linear layer, $L^+(x)$ denotes a linear layer with positive weights, and $s$ is a non-decreasing convex activation function.

To make $G$ strongly convex so that $\nabla G$ can be invertible, a quadratic term is added to $G(x)$ to obtain $G_\alpha(x) = \frac{\alpha}{2}\|x\|_2^2 + G(x)$, where $\alpha$ is a positive scalar such that $\nabla^2 G_\alpha \succeq \alpha I \succ 0$. This modification ensures that the gradient $\nabla G_\alpha$ is bijective onto its image (Huang et al., 2020) and the inverse can be easily computed via conjugation, which is a simple convex optimization. For the sake of brevity and clarity, we abuse the notation by omitting the sub-index $\alpha$, and will use $\nabla G$ to represent the CP-Flow model in the following of this paper.

As vanilla flow-based models, CP-Flow training centers on maximizing the log likelihood of the model density $\log \nabla G(x)$, whose derivative requires estimating the derivative of log determinant of the Hessian matrix for the convex potential in Equation 5 (Huang et al., 2020). By employing the Hutchinson trace estimator with a Rademacher random vector $v$ (Appendix C), the derivative of the log determinant of the Hessian matrix can be estimated as follows:

$$\frac{\partial}{\partial \vartheta} \log \det H = \mathbb{E}_v \left[ v^\top H^{-1} \frac{\partial H}{\partial \vartheta} v \right] \tag{6}$$

Evaluating the expression $v^\top H^{-1}$ in Equation 6 is prohibitively expensive. Instead, we treat it as the solution to a quadratic optimization problem $\arg\min_z \left\{ \frac{1}{2} z^\top H z - v^\top z \right\}$, given that $H$ is symmetric positive definite. In practice, one can use the conjugate gradient (CG) method to address the above quadratic optimization problem.

## 4 Algorithms

In this section, we begin by outlining the discretization method for mirror Langevin dynamics. Subsequently, we introduce our Convex Potential Mirror Langevin Algorithm (CPMLA), which uses CP-Flow as a dynamic mirror map for EBM sampling. The dynamic mirror map adaptively captures the underlying data distribution, facilitating flexible modifications in the sampling orientation according to the data distribution, thereby enhancing the algorithm's overall efficiency.

### 4.1 MIRROR LANGEVIN ALGORITHM

To generate synthesized examples from the probability distribution $p(x)$ with mirror Langevin dynamics (Hsieh et al., 2018), we need to solve the following equation

$$
\begin{aligned}
dY_t &= \nabla \log p(X_t)dt + \sqrt{2\nabla^2 G(X_t)}dW_t, \\
X_t &= \nabla G^*(Y_t)
\end{aligned}
\tag{7}
$$

where $W_t$ is the standard Brownian motion in $\mathbb{R}^d$ and $\nabla G$ is the mirror map. $\nabla G^*$ is the convex conjugate of $\nabla G$, which is defined by $G^*(y) = \max_{x \in \mathcal{K}}(\langle x, y \rangle - G(x))$. Appendix D provides a more approachable method for computing the inverse of the derivative of the mirror map, denoted as $(\nabla G)^{-1}$, which can be shown as $(\nabla G)^{-1} = \nabla G^*$.

In practical applications, we opt for the *Alternative Forward Discretization Scheme* ($\text{MLA}_{\text{AFD}}$) which has exponential convergence and vanishing bias (Jiang, 2021; Ahn & Chewi, 2021). Within this framework, we adopt the dynamic mirror map represented by $\nabla G$, as discussed in Section 3.2. It is pertinent to note that the computation of the gradient $\nabla G$ is typically less computationally demanding compared to the computation of $\nabla f$, which might entail a finite sum over a large number of data points. Specifically, at iteration $k$ with SDE step size $\eta$, we consider the following Equation 8.

$$
\begin{aligned}
x_{k+1/2} &\overset{1}{=} \nabla G^*(\nabla G(x_k) - \eta \nabla f(x_k)) \\
\text{solve } dy_t &= \sqrt{2\left[\nabla^2 G^*(y_t)\right]^{-1}}dW_t \\
&\overset{*}{=} \sqrt{2\nabla^2 G(y_t)}dW_t \text{ for } y_0 \overset{2}{=} \nabla G(x_{k+1/2}) \\
x_{k+1} &\overset{3}{=} \nabla G^*(y_\eta)
\end{aligned}
\tag{8}
$$

The $*$ step is derived from the property of convex conugate (Amari, 2016). The proof of this statement can be found in Appendix E.

The computations in Equation 8 can be simplified by noting that $\nabla G^* = (\nabla G)^{-1}$ (Appendix D). Specifically, $\nabla G^*$ in step 1 and $\nabla G$ in step 2, as well as $\nabla G$ in step 1 and $\nabla G^*$ in step 3, cancel out across iterations. The resulting simplified formulation is provided in Algorithm 1.

### 4.2 CPMLA

Our CPMLA employs $\text{MLA}_{\text{AFD}}$ as the algorithmic core and utilizes the aforementioned CP-Flow as a dynamic mirror map. CP-Flow adeptly transforms the geometry of data distribution with the underlying metric dictated by the Hessian of the convex potential. Consequently, the cooperation between $\text{MLA}_{\text{AFD}}$ and CP-Flow guides sampling on the data manifold, rendering it well-suited for fast convergence of the samples.

Like vanilla mirror Langevin algorithms, CPMLA follows an alternating sampling strategy between primal and dual spaces. The alternating sampling process entails transitioning between updating samples in the dual space using a dynamic mirror map for LMC exploration, followed by mapping the sample back to the primal space utilizing the inverse of the mirror map.

Specifically, at each iteration, for each batch $i = 1, \cdots, m$, we employ a two-step process within CPMLA. Initially, a noise vector $\xi$ is generated from a Gaussian distribution $\mathcal{N}(0, \nabla^2 G(x_i))$. This is a trick we use to avoid computing the square root of the Hessian, as $\sqrt{2\eta \nabla^2 G(x_i)} \cdot \tilde{\xi}$ where $\tilde{\xi} \sim \mathcal{N}(0, I)$ and $\sqrt{2\eta} \cdot \xi$ where $\xi \sim \mathcal{N}(0, \nabla^2 G(x_i))$ are equivalent. In practice, we only use the diagonal matrix of $\nabla^2 G$ for approximation. Additionally, noise examples $\{\hat{y}_i\}$ are generated from a standard Gaussian distribution $\mathcal{N}(0, I)$ in the dual space. From the set $\{\hat{y}_i\}$, we perform the standard EBM sampling steps in the dual space. These steps involve a gradient step and an SDE step, iterated for $T$ steps, yielding $\{y_i^{\text{EBM}}\}$. To obtain samples in the primal space, we employ the inverse of CP-Flow to map $\{y_i^{\text{EBM}}\}$ back, resulting in $\{x_i^{\text{out}}\}$. The synthesized examples $\{x_i^{\text{out}}\}$ are considered as outputs sampled by CPMLA.

Algorithm 1 provides a detailed description of the proposed CPMLA. As Section 4.1 stated, Algorithm 1 and Equation 8 may differ in form, but they are essentially the same. After the $k$-th iteration,

obtaining $x_{k+1}$ involves applying $\nabla G^*$ on $y_\eta$, and at the start of the $(k + 1)$-th iteration, applying $\nabla G$ again to $x_{k+1}$. Since $\nabla G^* = (\nabla G)^{-1}$ (Appendix D), these steps effectively cancel each other out, merely mapping between the primal and dual spaces. In Algorithm 1, the noise examples $\{\hat{y}_i\} \sim \mathcal{N}(0, I)$ inherently reside in the dual space, thus there is no need for another $\nabla G$. We can proceed directly with Langevin sampling.

---

**Algorithm 1** Convex Potential Mirror Langevin Algorithm (CPMLA)

---

**Input:** (1) Observed images $\{x_i\} \sim p_{\text{data}}(x)$; (2) Batch indicator $i = 1, \cdots, m$; (3) Number of Mirror Langevin steps $T$; (4) Number of steps in dual space $\eta$.
**Output:** Parameters of EBM and CP-Flow $\{\theta, \vartheta\}$
Randomly initialize $\theta$ and $\vartheta$.
**repeat**
    Sample noise vector $\xi \sim \mathcal{N}(0, \nabla^2 G(x_i))$ and noise examples $\{\hat{y}_i\} \sim \mathcal{N}(0, I)$ in dual space.
    **for** $k = 1 : T$ **do**
        Gradient descent part $\hat{y}_{i,k} = \hat{y}_{i,k} - \eta \nabla f(\hat{y}_{i,k})$
        SDE part $\hat{y}_{i,k} = \hat{y}_{i,k} + \sqrt{2\eta} \cdot \xi$
    **end for**
    Starting from $\{y_i^{\text{EBM}}\} \triangleq \{\hat{y}_{i,T}\}$, map back to primal space $x_i^{\text{out}} = \nabla G^*(y_i^{\text{EBM}})$
    Starting from $\{x_i^{\text{out}}\}$, update $\vartheta$ by Equation 6
    Given $\{x_i\}$ and $\{x_i^{\text{out}}\}$, update $\theta$ with Equation 3
**until** converged

---

For *cooperative learning* of EBM and CP-Flow, we update the parameters $\vartheta$ of the CP-Flow by feeding it with original examples $\{x_i\}$ and synthesized examples $\{x_i^{\text{out}}\}$ via CMPLA. Then, we update the parameters $\theta$ of the EBM, which is computed based on observed examples $\{x_i\}$ and synthesized examples $\{x_i^{\text{out}}\}$, as in Equation 3.

## 5 THEORETICAL ANALYSIS

In this section, we present the convergence proof for our CPMLA. Notably, we are the first to demonstrate the convergence of mirror Langevin algorithm for deep neural networks.

To ensure convergence, our proof incorporates inherent properties of neural networks, such as bounded gradients (or gradient clipping if unbounded). The following assumptions, detailed in Appendix F, are crucial for the analysis, and we further note that assumptions on neural networks are all satisfied within our specific setting.

**Assumption 5.1.** ($\beta$-Mirror Log-Sobolev Inequality, $\beta$-Mirror LSI) The target distribution $\pi$ satisfies $\beta$-Mirror LSI with constant w.r.t a given mirror map $\nabla G$, i.e., for every locally lipschitz function $h$, it holds that $\pi$ satisfies

$$\frac{2}{\beta} \int \|\nabla h\|^2_{[\nabla^2 G]^{-1}} d\pi \geq \int h^2 \log h^2 d\pi - \left( \int h^2 d\pi \right) \log \left( \int h^2 d\pi \right) \tag{9}$$

Log-Sobolev Inequality (LSI) is a well-known isoperimetric inequality that plays a crucial role in deriving the exponential concentration inequality using the entropy method.

**Assumption 5.2.** ($\zeta$-Self-Concordance) There exists a constant $\zeta \geq 0$ such that the conjugate mirror map $\nabla G^*$ satisfies that $\forall y, u, s, v$,

$$\left| \nabla^3 G^*(y)[u, s, v] \right| \leq 2\zeta \cdot \left( u^\top \nabla^2 G^*(y)u \right)^{1/2} \cdot \left( s^\top \nabla^2 G^*(y)s \right)^{1/2} \cdot \left( v^\top \nabla^2 G^*(y)v \right)^{1/2} \tag{10}$$

The self-concordance property is a crucial requirement in various barrier and entropy functions, such as the log-barrier function.

**Assumption 5.3.** ($L$-Relative Lipschitz) For all $x$, it holds that $f : \mathbb{R}^d \mapsto \mathbb{R}$ is differentiable with

$$\|\nabla f(x)\|_{[\nabla^2 G(x)]^{-1}} \leq L \tag{11}$$

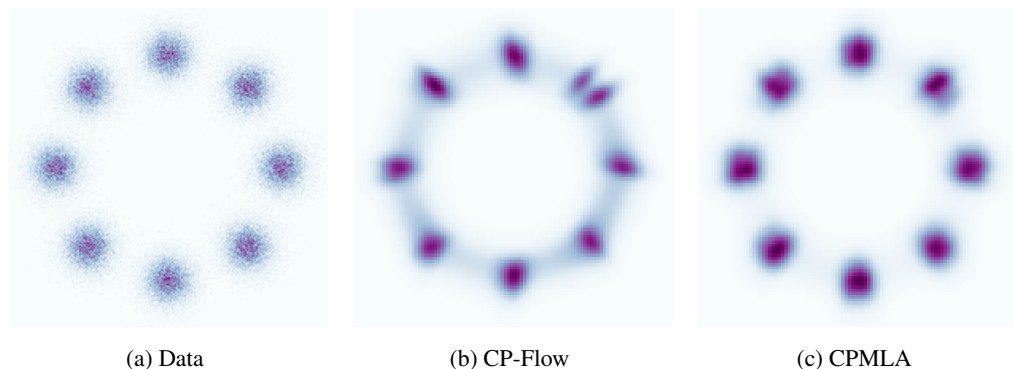

(a) Data            (b) CP-Flow            (c) CPMLA

Figure 1: Comparison between CPMLA and CP-Flow for Fitting Eight Gaussians. CPMLA reaches the same result in just 3 iterations that CP-Flow takes 10 iterations to achieve.

Assumption 5.3 extends the concept of Lipschitz continuity of a differentiable function, and has been useful to analyze discretizations of the mirror Langevin dynamics.

**Assumption 5.4.** (Weaker $\gamma$-Relative Smooth) For all $x, x' \in dom(G)$,

$$\|\nabla f(x) - \nabla f(x')\|_{[\nabla^2 G(x')]^{-1}} \le \gamma \cdot \|\nabla G(x) - \nabla G(x')\|_{[\nabla^2 G(x')]^{-1}} \tag{12}$$

This assumption can be viewed as generalizations of smooth functions (Lu et al., 2018).

**Theorem 5.5.** *Under Assumption 5.1 - 5.4, after* $k \ge \tilde{\Omega}\left(M\gamma^2 d/\beta^2\delta\right)$, *Algorithm 1 has an exponential convergence to the target distribution, that is* $\mathbb{D}_{KL}(\rho_k\|\pi) < \delta$.

**Proof sketch**: We begin with Lemma 1 from Jiang (2021), which provides us with the differential form of our algorithm in the primal space. This equation describes a weighted Langevin dynamics with a shifted drift term $\hat{\mu}$, which is crucial for understanding the convergence properties of the algorithm. We then consider the Fokker-Planck equation for the conditional density $\rho_{t|0}(x_t \mid x_0)$. Next, we focus on the KL-divergence between the time-dependent density $\rho_t$ and the target distribution $\pi$. Utilizing the integration by parts formula, Cauchy-Schwarz inequality and the mirror log-Sobolev inequality (Assumption 5.1) we obtain a differential inequality for the KL-divergence. This inequality shows that the KL-divergence decreases over time, with a rate determined by the algorithm's parameters and the properties of the target distribution.

The theorem states that, incorporating slight assumptions, CPMLA not only achieves exponential convergence but also exhibits vanishing bias. Details on this proof may be found in Appendix G.

## 6 EXPERIMENTS

We present a comprehensive evaluation of the proposed model on diverse tasks. To elucidate the fundamental concept of CPMLA, we commence with a toy example in Section 6.1. Next, we showcase compelling results for image generation in Section 6.2. Finally, we demonstrate the applicability of CPMLA in the realms of image reconstruction and inpainting in Section 6.3.

### 6.1 TOY MODEL STUDY

We first illustrate the efficacy of our approach using a toy example. Specifically, we apply CPMLA to model the density of eight Gaussian distributions, as described in Papamakarios et al. (2017) and Behrmann et al. (2019). The results, presented in Figure 1, clearly show that CPMLA efficiently fits these distributions. Notably, CPMLA achieves in 3 iterations what CP-Flow requires 10 iterations to reach (Huang et al., 2020), highlighting its superior convergence speed.

## 6.2 IMAGE GENERATION

We assess our model's performance using three distinct datasets for image synthesis: CIFAR-10 (Krizhevsky, 2009), which consists of 50,000 training images and 10,000 test images across 10 categories; SVHN (Netzer et al., 2011), a dataset with over 70,000 training images and more than 20,000 test images of house numbers; and CelebA (Liu et al., 2015), a large dataset of celebrity faces containing over 200,000 images. To ensure consistency and fair comparison, all images are downscaled to a resolution of $32 \times 32$ pixels before analysis. This standardized resolution facilitates fair comparisons across datasets and ensures a uniform approach to our synthesis evaluations. In our study, we present results for our model under two distinct settings, each offering valuable insights into its performance. CPMLA Setting: Here, both CP-Flow and EBM are trained from scratch; CPMLAprt Setting: In this setting, Initially, CP-Flow is pretrained using observed data. Subsequently, the CPMLA starts with the pretrained CP-Flow parameters. This setting can grasp an initial understanding of the data distribution and generate images with higher quality.

Through these scenarios, we present our findings combining both qualitative insights, as seen in Figure 3, and quantitative measures, detailed in Table 2. The FID scores, as proposed by Heusel et al. (2017), are computed based on 50,000 samples from each dataset. Our models exhibit superior performance over most baseline algorithms, achieving lower FID scores compared to standalone normalizing flows and outperforming prior studies that simultaneously train a normalizing flow with an EBM (Gao et al., 2019; Nijkamp et al., 2020). Our results are comparable to those of leading EBMs.

In particular, compared to CoopFlow, CPMLA provides a distinct advantage in terms of inference efficiency. **(i)** Table 2 indicates that **CPMLA can achieve a FID score comparable to CoopFlow with fewer iterations of LMC.** At $T = 20$, CPMLA reaches a FID of 21.43, which is a 30.29% reduction compared to CoopFlow's FID of 30.74 at the same iteration. Moreover, CPMLA's FID at $T = 20$ is comparable to CoopFlow's FID at $T = 30$. Figure 2 shows changes of FID of these two models from $T = 3$ to $T = 21$, further demonstrating the superior inference speed of CPMLA. **(ii)** Furthermore, CP-Flow, the component utilized by CPMLA as initialization, exhibits a lower parameter amount (0.27M) compared to the normalizing flow (28.78M) utilized by CoopFlow, as shown in Table 1. This means that CPMLA achieves comparable performance to CoopFlow using **only 0.9% of its parameter count**.

It is important to note that NCSN++, a score-based SDE model, is fundamentally different from CPMLA, making direct comparisons between the two inappropriate. CPMLA demonstrates advantages in terms of modeling simplicity and training stability which makes it faster to train and more easily implemented.

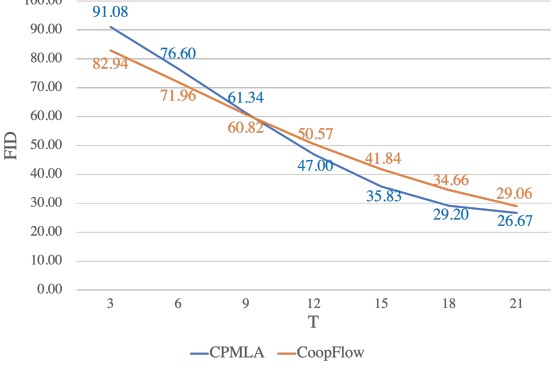

|  | EBM part | Flow part |
|---|---|---|
| CoopFlow | 17.13M | 28.78M |
| **CPMLA** | 17.13M | 0.26M |

Table 1: Comparison of the parameter amount between CoopFlow and CPMLA. CPMLA achieves comparable FID scores to CoopFlow with only 0.9% parameter count.

Figure 2: FID comparison from $T = 3$ to $T = 21$ between CPMLA and CoopFlow on CIFAR-10 dataset. From an inferior initialization, CPMLA demonstrates faster inference speeds than CoopFlow.

| Model type | Models | FID↓ |
|---|---|---|
| VAE | VAE (Kingma & Welling, 2013) | 78.41 |
| Autoregressive | PixelCNN (Salimans et al., 2017) | 65.93 |
| GAN | WGAN-GP (Gulrajani et al., 2017) | 36.40 |
| | StyleGAN2-ADA (Karras et al., 2020) | 2.92 |
| Score-Based | NCSN (Song & Ermon, 2019) | 25.32 |
| | NCSN++ (Song et al., 2020) | 2.20 |
| Flow | Glow (Kingma & Dhariwal, 2018) | 45.99 |
| | Residual Flow (Chen et al., 2019) | 46.37 |
| EBMs | LP-EBM (Pang et al., 2020) | 70.15 |
| | EBM-SR (Nijkamp et al., 2019) | 44.50 |
| | EBM-IG (Du & Mordatch, 2019) | 38.20 |
| | CoopVAEBM (Xie et al., 2021) | 36.20 |
| | CoopNets (Xie et al., 2020) | 33.61 |
| Flow+EBM | NT-EBM (Nijkamp et al., 2020) | 78.12 |
| | EBN-FCE (Gao et al., 2019) | 37.30 |
| | CoopFlow (T=20) | 30.74 |
| | CoopFlow (T=30) (Xie et al., 2022b) | 21.16 |
| **CPMLA** (Ours) | CPMLAprt (T=20) | 21.43 |
| | CPMLA (T=20) | 26.67 |

Table 2: FID scores on the CIFAR-10. Our CPMLA demonstrates superior efficiency in EBMs sampling. Other generative models may have a lower FID, but they are not included in the direct comparison with our method.

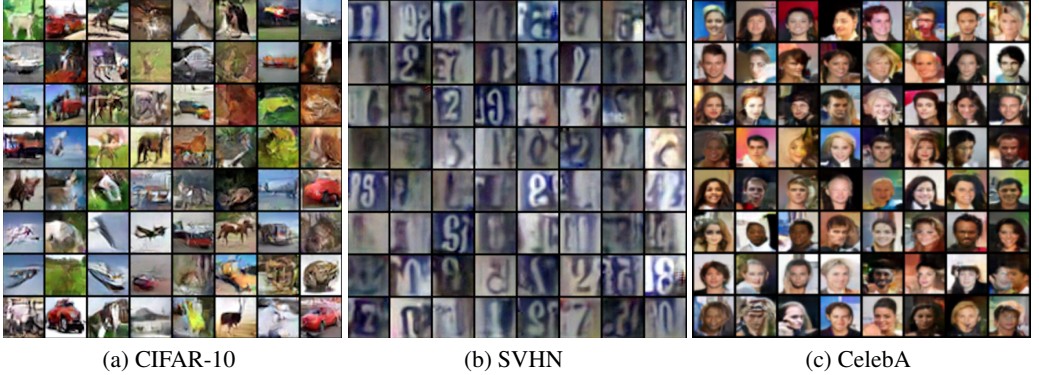

      (a) CIFAR-10              (b) SVHN              (c) CelebA

Figure 3: Generated Samples (32 × 32 pixels) by CPMLA from CIFAR-10, SVHN, and CelebA datasets. These images are produced under the CPMLAprt training setting.

## 6.3 IMAGE RECONSTRUCTION AND INPAINTING

We demonstrate the efficiency and effectiveness of CPMLA for image reconstruction task, with a focus on the CIFAR-10 testing set as illustrated in Figure 4 (Appendix I). The fidelity of the reconstructed images to the originals is a testament to the model's capability. This empirical evidence solidifies the stance that CPMLA framework functions effectively as a latent variable model, thereby confirming its theoretical underpinnings and practical applicability in image reconstruction tasks.

We further demonstrate the versatility of CPMLA in the context of image inpainting. Let's assume we have an image represented by a function $I : \Omega \subset \mathbb{R}^2 \to \mathbb{R}^3$, where $\Omega$ is the domain of the image, and $I(x, y)$ gives the color at coordinates $(x, y)$. We optimize the objective energy function:

$$E(u) = \int_\Omega (I(x, y) - u(x, y))^2 \cdot M(x, y) dx dy \tag{13}$$

to measure the difference between the restored region and the original image. To ensure that the restoration process does not alter the undamaged parts of the original image, we introduce a constraint: $u(x, y) = I(x, y)$ if $M(x, y) = 1$. Our experiments, conducted on the CelebA training set, are encapsulated in Figure 5 (Appendix I). The first 17 columns exhibit the inpainting results at various optimization iterations, offering a dynamic view of the reconstruction process. The last two columns visually compare the masked images and the originals. Remarkably, Figure 5 illustrates that CPMLA excels in faithfully reconstructing and inpainting the masked images through diverse initializations.

## 7 LIMITATIONS AND FUTURE WORKS

In our experiments, estimating the Hessian can introduce bias to the optimal point and incur additional computational cost. However, compared to the exact evaluation of the inverse Hessian, this is a trade-off we must make. While our experimental results demonstrate effectiveness for diverse sampling tasks, the assumption of the log-Sobolev inequality (Assumption 5.1) is rather general, as we cannot ensure that the target distributions of different sampling tasks satisfy this assumption, particularly in EBMs where the target distribution is highly complex. We must point out that this paper provides improvement in the quality of generative models could be used to generate deepfakes for disinformation. For future work, we plan to explore the deeper connection between sampling and optimization. For instance, are various optimization tricks, such as Trust-Region, RMSProp and Adam, justified for either accelerating sampling or correcting for bias? Additionally, would higher-order discretization schemes, such as Runge-Kutta methods, yield better convergence rates? We aim to investigate these questions and further advance the field of sampling and optimization.

## 8 CONCLUSIONS

In this paper, we propose a novel approach CPMLA, an efficient sampling method for EBMs. Its ability to adapt to the geometry of data manifolds through the utilization of CP-Flow as the dynamic mirror map enables efficient and no-bias convergence. The algorithm's theoretical underpinnings and practical applicability are further solidified by its performance in image generation, reconstruction, and inpainting tasks, showcasing its potential for sampling accuracy and inference efficiency in real-world applications.

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

## A   Optimal Transport

In recent years, there has been increasing interest in applying optimal transport theory to generative modeling, which considers the training process as a task of minimizing the distance between two probability distributions. More specifically, the objective is to transform a random distribution into a target distribution that closely approximates the underlying data distribution, with the distance between these two distributions often quantified using the Wasserstein distance in the context of optimal transport. The Wasserstein $p$-distance between two probability measures $\mu$ and $\nu$ on a metric space $M$ with finite $p$-moments is

$$W_p(\mu, \nu) = \left( \inf_{\gamma \in \Gamma(\mu, \nu)} \mathbf{E}_{(x,y) \sim \gamma} d(x, y)^p \right)^{1/p} \tag{14}$$

where $\Gamma(\mu, \nu)$ is the set of all couplings of $\mu$ and $\nu$. Jordan et al. (1998) shows that the Langevin dynamics manifests as the gradient flow of the Kullback-Leibler divergence within the probability measure space, characterized by the Wasserstein metric, as elucidated through the Fokker-Planck equation. This observation establishes a more substantial linkage between the realms of sampling and optimization; see also by the paper Otto & Villani (2000).

## B   Mirror Langevin Algorithm for Constrained Sampling

The mirror Langevin algorithm is a powerful technique for sampling from complex distributions, particularly those with constraints or intricate geometries. It leverages the concept of mirror maps, which can adapt to the geometric structure of the target distribution, enabling efficient and accurate sampling. One notable application of the mirror Langevin algorithm is constrained sampling, where the goal is to draw samples from a population while adhering to specific conditions or constraints.

Constrained sampling involves drawing a set of samples $S$ from a population $U = u_1, u_2, ..., u_n$ while satisfying predefined constraint conditions expressed as inequalities. The general form of these constraints can be written as:

$$C(x) : g_i(x) \leq 0, \quad i = 1, 2, ..., m \tag{15}$$

Here, $g_i(x)$ represents the constraint functions, and the goal is to ensure that $g_i(x) \leq 0$ for all $i$.

The mirror Langevin algorithm addresses constrained sampling by leveraging mirror maps that can adapt to the geometry of the constraints. Specifically, we employ CP-Flow as our dynamic mirror map, which utilizes Implicit Convex Neural Networks (ICNNs) to approximate arbitrary convex functions effectively. Given that the derivative of a convex function is monotonic, we can ensure that the convergence of potential functions implies the convergence of the associated gradient fields, as stated in the following theorem:

**Theorem B.1** (Optimality (Theorem 4 in Huang et al. (2020))). *Let $F$ be the Brenier potential of $X \sim \mu$ and $Y \sim \nu$, and let $G_n$ be a convergent sequence of differentiable, convex potentials, such that $\nabla G_n \circ X \to Y$ in distribution. Then, $\nabla G_n$ converges almost surely to $\nabla F$.*

In our context, where $\mu$ represents the unconstrained space and $\nu$ serves as the convex constraint, Theorem B.1 guarantees the existence of a convex potential whose derivative maps $\mu$ to $\nu$. By leveraging the expressive nature of ICNNs through CP-Flow, we can adapt to the intrinsic geometry of the constraints, resulting in accelerated convergence during constrained sampling. The mirror Langevin algorithm's ability to handle complex constraints makes it a valuable tool for various applications beyond constrained sampling, such as sampling from EBMs, Bayesian inference, and more (Ahn & Chewi, 2021).

## C   Details of Training CP-Flow

Huang et al. (2020) presents an alternative formulation of the gradient as the solution to a convex optimization problem, eliminating the need to differentiate through the log-determinant estimation process. By adapting the gradient formula from Appendix C in Chen et al. (2019) to the context of convex potentials, and utilizing Jacobi's formula* alongside the adjugate representation of the matrix inverse †, we derive the following identity for any invertible matrix $H$ parameterized by $\theta$:

$$\frac{\partial}{\partial\theta}\log\det H = \frac{1}{\det H}\frac{\partial}{\partial\theta}\det H \overset{*}{=} \frac{1}{\det H}\operatorname{tr}\left(\operatorname{adj}(H)\frac{\partial H}{\partial\theta}\right) \overset{\ddagger}{=} \operatorname{tr}\left(H^{-1}\frac{\partial H}{\partial\theta}\right) = \mathbb{E}_v\left[v^\top H^{-1}\frac{\partial H}{\partial\theta}v\right]$$

(16)

In the last equality, Huang et al. (2020) apply the Hutchinson trace estimator using a Rademacher random vector $v$, which is an unbiased Monte Carlo gradient estimator.

## D  PROPERTY OF CONVEX CONJUGATE

$G^*$ is the convex conjugate of $G$. Then

$$\nabla G(x) = x^*(x) := \arg\sup_{x^*}\langle x, x^*\rangle - G^*(x^*)$$
$$\nabla G^*(x^*) = x(x^*) := \arg\sup_{x}\langle x, x^*\rangle - G(x)$$

(17)

Hence

$$x = \nabla G^*(\nabla G(x)) \quad\text{and}\quad x^* = \nabla G(\nabla G^*(x^*))$$

(18)

## E  LEMMA OF CONVEX CONJUGATE

**Lemma E.1.**  *Suppose we have a dualistic structure*

$$\boldsymbol{\xi}^* = \nabla G(\boldsymbol{\xi}), \quad \boldsymbol{\xi} = \nabla G^*(\boldsymbol{\xi}^*)$$

(19)

*$G^*$ is the Legendre dual of $G$, which is defined as*

$$G^*(\boldsymbol{\xi}^*) = \max_{\boldsymbol{\xi}'}\{\boldsymbol{\xi}'\cdot\boldsymbol{\xi}^* - G(\boldsymbol{\xi}')\}$$

(20)

*Then the Hessian of $G^*(\boldsymbol{\xi}^*)$ is written as*

$$\nabla\nabla G^*(\boldsymbol{\xi}^*) = \frac{\partial\boldsymbol{\xi}}{\partial\boldsymbol{\xi}^*}$$

(21)

*which is the inverse of the Hessian of $G(\boldsymbol{\xi})$*

$$\frac{\partial\boldsymbol{\xi}^*}{\partial\boldsymbol{\xi}} = \nabla\nabla G(\boldsymbol{\xi})$$

The last step is guaranteed by $\nabla G^* = \nabla G^{-1}$, which can be shown from Appendix D.

## F  DETAILS OF ASSUMPTIONS

Previous investigations into the mirror Langevin algorithm (Zhang et al., 2020) have required the relative $\mu$-strong convexity of $f$ with respect to $G$ to guarantee convergence. However, our work introduces Assumption 5.1, which relaxes this requirement and permits consideration of non-strongly convex distributions. Assumption 5.1 can be transformed as following. Taking $h(x) = \sqrt{\frac{d\rho(x)}{d\pi(x)}}$, then $\forall\rho$

$$\mathbb{D}_{KL}(\rho\|\pi) := \int \rho(x)\log\frac{\rho(x)}{\pi(x)}dx \leq \frac{1}{2\beta}\int \rho(x)\left\|\nabla\log\frac{\rho(x)}{\pi(x)}\right\|^2_{[\nabla^2 G(x)]^{-1}}dx =: \frac{1}{2\beta}J^G_\pi(\rho) \quad (22)$$

The $\mathbb{D}_{KL}(\rho\|\pi)$ term represents the KL divergence, often serving as a measure of the distance between distributions $\rho$ and $\pi$. On the other hand, the right-hand side term, $J^G_\pi(\rho)$, signifies the weighted Fisher information. As demonstrated by Jordan et al. (1998), Langevin dynamics can be interpreted as the gradient flow of the KL divergence within the space of probability measures, equipped with the Wasserstein metric through the Fokker-Planck equation. This connection establishes a link between

sampling and optimization.In this context, Assumption 5.1 can be perceived as the condition of gradient domination for KL-divergence in the Wasserstein metric.

Assumption 5.2 specifically relates to the interplay between the higher-order derivatives and the lower-order derivatives of the function. When the secondary derivative is small, it implies that the first derivative, which is governed by the secondary derivative, is also small. This property ensures the solution of continuous dynamics and Hessian stability (Zhang et al., 2020), indicating that the underlying geometry does not undergo rapid changes. Moreover, this property is preserved under Fenchel conjugation (with the same parameter), affine transformation and summation (Nesterov & Nemirovskii, 1994). The concept of self-concordance is also prevalent in quadratic optimizations, such as the interior point method, where it guarantees the convergence performance $O\left(\sqrt{\zeta}\log\frac{1}{\epsilon}\right)$.

In Assumption 5.3, when $G(x)=\frac{\|x\|^2}{2}$, we regain the conventional definition of a differentiable function being Lipschitz continuous with a parameter $\beta$. This property has been extensively employed in prior research. In the case where $G=f$ and a function $f$ satisfies $\|\nabla f(x)\|_{[\nabla^2 f(x)]^{-1}} \leq L$, it is referred to as a barrier function (Nesterov et al., 2018). This property also emerges in the analysis of Newton's method in quadratic optimization scenarios.

In formal algorithms, it is often necessary to have the $\gamma$-relative smooth property in order to ensure convergence. $\gamma$-relative smooth is defined by

$$
\begin{aligned}
&\left\|\left[\nabla^2 G(x)\right]^{-1}\nabla f(x)-\left[\nabla^2 G\left(x'\right)\right]^{-1}\nabla f\left(x'\right)\right\|_{\nabla^2 G(x')} \\
&\leq \gamma \cdot \left\|\nabla G(x)-\nabla G\left(x'\right)\right\|_{[\nabla^2 G(x')]^{-1}}
\end{aligned}
\tag{23}
$$

However, the CPMLA utilizes a distinct approach by employing deterministic gradient steps and stochastic steps separately. This allows for the utilization of a weaker notion of smoothness assumption, namely Assumption 5.4. Unlike the $\gamma$-relative smooth, which necessitated Lipschitz continuity across different metrics $\nabla^2 G$ and could be unavoidable when discretizing the geometry, this definition of relative smoothness only considers the local metric $\nabla^2 G$ at a single point.

## G   PROOF OF THEOREM 5.5

*Proof.* Lemma 1 in Jiang (2021) tells us that the differential form of Algorithm 1 in primal space is

$$
\begin{aligned}
dX_t =& -\left[\nabla^2 G\left(X_t\right)\right]^{-1}\nabla f\left(X_0\right)dt - \left[\nabla^2 G\left(X_t\right)\right]^{-1}\operatorname{Tr}\left(\nabla^3 G\left(X_t\right)\left[\nabla^2 G\left(X_t\right)\right]^{-1}\right)dt \\
& + \sqrt{2\left[\nabla^2 G\left(X_t\right)\right]^{-1}}dW_t \\
=& \left[-\left[\nabla^2 G(x_t)\right]^{-1}\nabla f(x_0) + \left[\nabla^2 G(x_t)\right]^{-1}\nabla f(x_t) - \left[\nabla^2 G(x_t)\right]^{-1}\nabla f(x_t)\right. \\
& \left. - \left[\nabla^2 G\left(X_t\right)\right]^{-1}\operatorname{Tr}\left(\nabla^3 G\left(X_t\right)\left[\nabla^2 G\left(X_t\right)\right]^{-1}\right)\right]dt + \sqrt{2\left[\nabla^2 G\left(X_t\right)\right]^{-1}}dW_t \\
=& \left(\nabla \cdot H^{-1}\left(X_t\right) - H^{-1}\left(X_t\right)\nabla f\left(X_t\right) + \hat{\mu}\right)dt + \sqrt{2H^{-1}\left(X_t\right)}dW_t
\end{aligned}
\tag{24}
$$

where we denote $\hat{\mu}=\left[\nabla^2 G\left(X_t\right)\right]^{-1}\left(\nabla f\left(X_t\right)-\nabla f\left(X_0\right)\right)$ and $H^{-1}=[\nabla^2 G]^{-1}$.

This is a weighted Langevin dynamics with shifted drift $\hat{\mu}$ (the reason of the convergence to a biased limit).

Now consider the Fokker-Planck equation for the conditional density $\rho_{t|0}\left(x_t \mid x_0\right)$. For the drift $b = \nabla \cdot H^{-1} - H^{-1}\nabla f + \hat{\mu}$, applying Lemma 3 in Wibisono (2019), we have

$$\frac{\partial \rho_t(x)}{\partial t} = \int \frac{\partial \rho_{t|0}\left(x \mid x_0\right)}{\partial t} \rho_0\left(x_0\right) dx_0$$

$$= \int \left[ -\nabla \cdot \left( \rho_{t|0}\left( \nabla \cdot G_0(x) - G_0(x)\nabla f(x) \right) \right) + \left\langle \nabla^2, \rho_{t|0}G_0(x) \right\rangle - \nabla \cdot \left( \rho_{t|0}\hat{\mu}_0(x) \right) \right] \rho_0\left(x_0\right) dx_0$$

$$= \nabla \cdot \left( \rho_{0|t} \int -\left( \rho_t \left( \nabla \cdot G_0(x) - G_0(x)\nabla f(x) \right) \right) + \nabla \cdot \left( \rho_t G_0(x) \right) dx_0 \right) - \nabla \cdot \left( \rho_t \int \rho_{0|t}\hat{\mu}_0(x) dx_0 \right)$$

$$= \nabla \cdot \left( \rho_{0|t} \int G_0(x)\nabla \rho_t + \rho_t G_0(x)\nabla f(x) dx_0 \right) - \nabla \cdot \left( \rho_t \int \rho_{0|t}\hat{\mu}_0(x) dx_0 \right)$$

$$= \nabla \cdot \left( \rho_{0|t} \int \left( \rho_t G_0(x)\nabla \log \frac{\rho_t}{\pi(x)} \right) dx_0 \right) - \nabla \cdot \left( \rho_t \int \rho_{0|t}\hat{\mu}_0(x) dx_0 \right) \tag{25}$$

where the last equality is because $\nabla \log \frac{\rho}{\pi} = \nabla(\log \rho + f)$.

Now consider the KL-divergence

$$\frac{d}{dt}\mathbb{D}_{KL}(\rho_t \| \pi) = \int \frac{d\rho_t}{dt} \log \frac{\rho_t}{\pi} dx + \int \pi \frac{1}{\pi}\frac{d\rho_t}{dt} dx = \int \frac{d\rho_t}{dt} \log \frac{\rho_t}{\pi} dx \tag{26}$$

According to Equation 25, we have

$$\frac{d}{dt}\mathbb{D}_{KL}(\rho_t \| \pi) = \int \frac{d\rho_t}{dt} \log \frac{\rho_t}{\pi} dx$$

$$= \int \nabla \cdot \left( \rho_{0|t} \int \rho_t G_0 \nabla \log \frac{\rho_t}{\pi(x)} dx_0 \right) \log \frac{\rho_t}{\pi} dx - \int \nabla \cdot \left( \rho_t \int \rho_{0|t}\hat{\mu}_0 dx_0 \right) \log \frac{\rho_t}{\pi} dx$$

$$= -\int \rho_{0|t} \int \rho_t \left\langle \nabla \log \frac{\rho_t}{\pi} G_0, \nabla \log \frac{\rho_t}{\pi} \right\rangle dx_0 dx + \int \rho_t \int \rho_{0|t} \left\langle \hat{\mu}_0, \nabla \log \frac{\rho_t}{\pi} \right\rangle dx_0 dx$$

$$= -\mathbb{E}_{\rho_t} \left[ \left\| \nabla \log \frac{\rho_t}{\pi} \right\|^2_{H^{-1}} \right] + \mathbb{E}_{\rho_{0,t}} \left[ \left\langle \hat{\mu}, \nabla \log \frac{\rho_t}{\pi} \right\rangle \right]$$

$$\leq -\mathbb{E}_{\rho_t} \left[ \left\| \nabla \log \frac{\rho_t}{\pi} \right\|^2_{[\nabla^2 G]^{-1}} \right] + \mathbb{E}_{\rho_{0,t}} \left[ \|\hat{\mu}\|^2_{\nabla^2 G} \right] + \frac{1}{4}\mathbb{E}_{\rho_t} \left[ \left\| \nabla \log \frac{\rho_t}{\pi} \right\|^2_{[\nabla^2 G]^{-1}} \right]$$

$$\leq -\frac{3\beta}{2}\mathbb{D}_{KL}(\rho_t \| \pi) + \mathbb{E}_{\rho_{0,t}} \left[ \|\hat{\mu}\|^2_{\nabla^2 G} \right] \tag{27}$$

The third equality refers to the integration by parts formula $\int \langle \nabla G(x), v(x) \rangle dx = -\int G(x)\nabla \cdot v(x)dx$. The first inequality is because $x^\top y \leq \|x\|_2^2 + \frac{1}{4}\|y\|_2^2$ and the last inequality is from Mirror LSI (Assumption 5.1).

Under Assumption 5.2 - 5.4, $\|\nabla G\|^2 \leq M_G$ are naturally satisfied in our setting. We have

$$\mathbb{E}_{\rho_{0,t}} \left[ \|\hat{\mu}\|^2_{\nabla^2 G} \right] \leq \gamma^2 \cdot \mathbb{E}_{\rho_{0,t}} \left[ \|\nabla G\left(x_t\right) - \nabla G\left(x_0\right)\|^2_{[\nabla^2 G(x_t)]^{-1}} \right]$$

$$= \gamma^2 \cdot \mathbb{E} \left[ \left\| -t\nabla f\left(x_0\right) + \sqrt{2}\int_0^t \left[ \nabla^2 G\left(x_s\right) \right]^{1/2} dW_s \right\|^2_{[\nabla^2 G(x_t)]^{-1}} \right] \tag{28}$$

$$\leq 2\gamma^2 t^2 \mathbb{E} \|\nabla f\left(x_0\right)\|^2_{[\nabla^2 G(x_t)]^{-1}} + 4\mathbb{E} \int_0^t \|\nabla^2 G\left(x_s\right)\|_{[\nabla^2 G(x_t)]^{-1}} ds$$

$$\leq 2\gamma^2 t^2 L^2 + 4t\gamma^2 M_G d$$

where the second inequality we use Itô isometry and $(a+b)^2 \leq 2(a^2+b^2)$.

Then if $0 \leq t \leq h$, we have

$$\frac{d}{dt}\mathbb{D}_{KL}(\rho_t \| \pi) \leq -\frac{3\beta}{2}\mathbb{D}_{KL}(\rho_t \| \pi) + 2\gamma^2 h^2 L^2 + 4h\gamma^2 M_G d \tag{29}$$

which is

$$\frac{d}{dt}\left(e^{\frac{3\beta}{2}t}\mathbb{D}_{KL}(\rho_t\|\pi)\right) \le e^{\frac{3\beta}{2}t}\left(2\gamma^2 h^2 L^2 + 4h\gamma^2 M_G d\right) \tag{30}$$

Integrate it for $0 \le t \le h$,

$$e^{\frac{3\beta}{2}h}\mathbb{D}_{KL}(\rho_h\|\pi) - \mathbb{D}_{KL}(\rho_0\|\pi) \le \frac{2}{3\beta}\left(e^{\frac{3\beta h}{2}} - 1\right)\left(2\gamma^2 h^2 L^2 + 4h\gamma^2 M_G d\right) \tag{31}$$

Then

$$\mathbb{D}_{KL}(\rho_h\|\pi) \le e^{-\frac{3\beta}{2}h}\mathbb{D}_{KL}(\rho_0\|\pi) + \frac{2}{3\beta}(1 - e^{-\frac{3\beta h}{2}})\left(2\gamma^2 h^2 L^2 + 4h\gamma^2 M_G d\right) \tag{32}$$

Iterating the recursion,

$$\mathbb{D}_{KL}(\rho_k\|\pi) \le e^{-\frac{3\beta}{2}hk}\mathbb{D}_{KL}(\rho_0\|\pi) + \frac{2}{3\beta}\left(2\gamma^2 h^2 L^2 + 4h\gamma^2 M_G d\right) \tag{33}$$

Using Lemma 6 in Jiang (2021) for initialization, picking the assumed stepsize, after $k \ge \tilde{\Omega}\left(M\gamma^2 d/\beta^2\delta\right)$, we have $\mathbb{D}_{KL}(\rho_t\|\pi) < \delta$. $\qquad\square$

## H   EXPERIMENTAL DETAILS

| Parameter | Value |
|-----------|-------|
| Dataset Size | 50,000 samples |
| Dynamic Mirror Map $\nabla G$ | 1 CP-Flow block |
| Depth | 20 |
| Dimh | 32 |
| $\nabla G$ Optimizer | Adam |
| $\nabla G$ Activation | Gaussian Softplus |
| $\nabla G$ Initial Learning Rate | 0.005 |
| EBM $\nabla f$ | 4 linear layers |
| $\nabla f$ Optimizer | Adam |
| $\nabla f$ Activation | Swish |
| $\nabla f$ Initial Learning Rate | 0.005 |
| Batch Size | 128 |
| Reported Results | After 3 and 10 epochs |

Table 3: Experimental setup for toy dataset

Table 3 outlines the experimental setup for an Eight Gaussian toy dataset experiment. This setup includes a dataset size of 50,000 samples, using single CP-Flow block with the Gaussian Softplus activation function for the Dynamic Mirror Map $\nabla G$. The optimizer for $\nabla G$ is Adam, with a and an initial learning rate of 0.005. The EBM $\nabla f$ comprises four linear layers with Swish activation, also utilizing the Adam optimizer, and an initial learning rate of 0.005. The batch size for this setup is 128, with reported results after 3 and 10 epochs.

Table 4 presents the experimental setup for the CIFAR-10, SVHN, and CelebA datasets. We use a multi-scale structure, involving two CP-Flow blocks, followed by invertible downsampling, and then another two CP-Flow blocks. All ICNN architectures had two hidden layers. For the EBM in CPMLA, a 3 blocks network is used to design the negative energy function. The following Table 5 presents training time of our model on each dataset on eight 3090 GPUs.

In the CPMLAprt setting, we first pretrain a CP-Flow on training examples, and then train a 30-step mirror Langevin sampling, whose parameters are initialized randomly, together with the pretrained CP-Flow by following Algorithm 1.

| Parameter | Value |
|---|---|
| Datasets | CIFAR-10, SVHN, CelebA |
| Dynamic Mirror Map $\nabla G$ | 2 CP-Flow blocks |
| ICNN Architecture | 2 hidden layers |
| $\nabla G$ Optimizer | Adam |
| $\nabla G$ Activation | Gaussian Softplus |
| $\nabla G$ Initial Learning Rate | 5e-4 |
| $\nabla G$ Weight Decay | 5e-5 |
| Mirror Steps | 10 |
| Mirror Step Size | 1e-2 |
| EBM $\nabla f$ | 3 blocks with 3 convolutional layers |
| $\nabla f$ Optimizer | Adam |
| $\nabla f$ Activation | Swish |
| $\nabla f$ Initial Learning Rate | 5e-3 |
| Batch Size | 128 |
| Reported Results | After 200 epochs |

Table 4: Experimental setup for CIFAR-10, SVHN, and CelebA datasets

| | CIFAR-10 | SVHN | CelebA |
|---|---|---|---|
| Training time (hours) | 18.5 | 22.2 | 70.9 |

Table 5: Training time on each dataset on eight 3090 GPUs

# I   MORE IMAGE GENERATION RESULTS

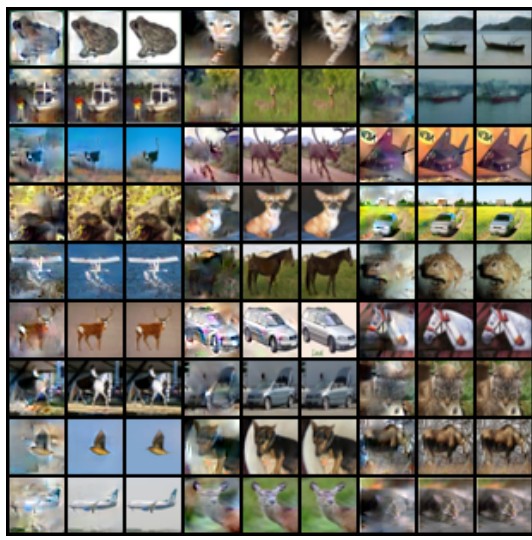

Figure 4: Image reconstruction on the CIFAR-10. The right column showcases the original images. The left and middle columns feature flow-generated images and the reconstructed images, respectively. We can see that the reconstruction is almost the same as the original one, which solidifies the stance that CPMLA functions effectively as a sampling algorithm.

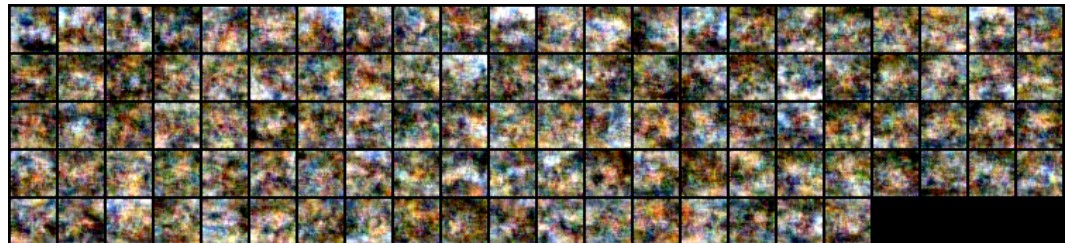

Figure 5: Image inpainting on the CelebA. The first 17 columns exhibit the inpainting results at various iterations, while the last two columns visually compare the masked images and the originals. CPMLA faithfully inpaints the masked images.

In Section 6.2, we have shown generated examples from CPMLA. In this section, we first show the examples generated by LMC on CIFAR-10. Then we compare the examples generated by the CP-Flow only and CPMLAprt in Figure 7 and 8. We can see huge difference between two algorithms and our generated examples are meaningful.

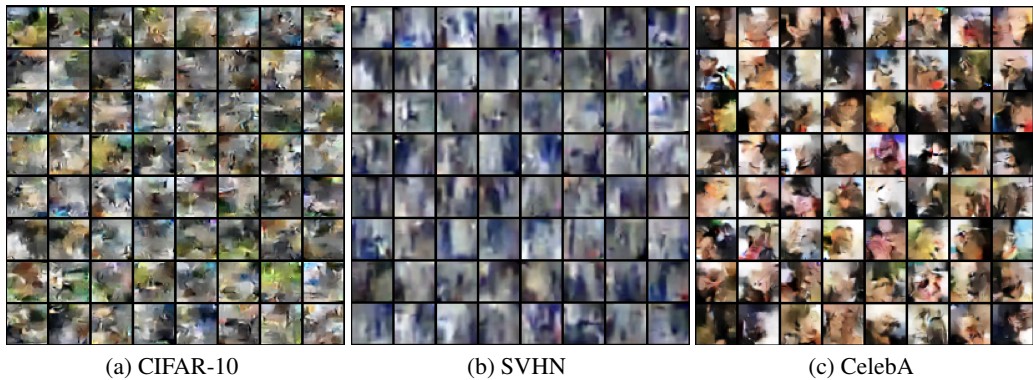

Figure 6: LMC on CIFAR-10

| (a) CIFAR-10 | (b) SVHN | (c) CelebA |

Figure 7: CP-Flow results

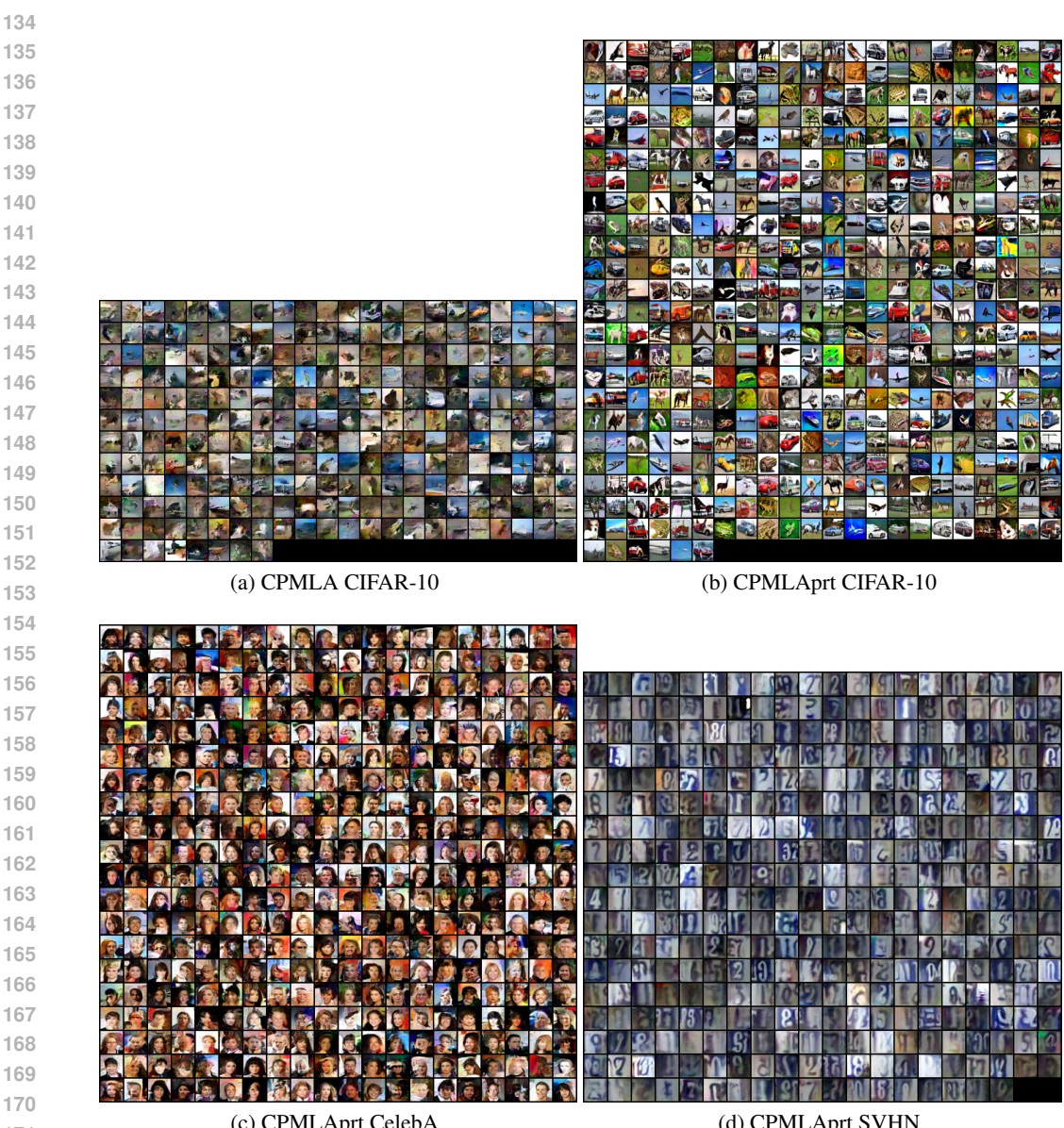

(a) CPMLA CIFAR-10                    (b) CPMLAprt CIFAR-10

(c) CPMLAprt CelebA                    (d) CPMLAprt SVHN

Figure 8: CPMLAprt results

