# OpenReview forum: "Convex Potential Mirror Langevin Algorithm for Efficient Sampling of Energy-Based Models"
_ICLR.cc/2025/Conference — Submitted to ICLR 2025_

### Official Review · Reviewer_67JH · 2024-10-26

**Soundness:** 1
**Presentation:** 2
**Contribution:** 2
**Rating:** 3
**Confidence:** 3

**Summary:**

The authors consider the specific sampling problem where the potential is given by some learned energy. This work proposes learning a convex mirror potential alongside the energy-based model for generation, motivated by the optimal mappings given by Brenier's theorem. The authors make two key approximations to computing the convex conjugate of an input-convex neural network, and in computing the Hessian of the optimal mapping. The sampling algorithm boils down to constructing a dual mapping, performing mirror Langevin sampling in the dual space from dual noise, and mirroring back to the primal space.

**Strengths:**

* The idea of jointly training an energy model along with another convex model to guide training seems new. There seems to be a hidden adversarial learning perspective, where the convex guide model is learned to have large deviation between clean and generated samples, which allows for an intuitive interpretation as "larger gradient steps towards ground truth samples".

**Weaknesses:**

* (Critical) The algorithm and corresponding theory in its current form are dubious. There is no mention as to how the inverse map $\nabla G^*$ is computed in practice. The theory claims to show convergence of the CPMLA algorithm *during training*, but does not consider the learnable parameters (there is no mention of $\theta$ or $\vartheta$ anywhere), and appears to be a clone of Prop. 3 in Jiang (2021). I suspect it might be possible using techniques from De Bortoli et al. (2021), but this is not considered in this work.

* (Critical) Algorithmically, there seems to be a critical typo in Algorithm 1 on line 284, on the "gradient descent part", where $\nabla f$ is evaluated on an element of the dual space. This appears also in Equation 8 in the starred equality: it should be $\nabla G(\nabla G^* y_t)$ inside the sqrt. This raises some suspicion about how the authors performed the experiments.
* There is a lack of explanation of how the convex conjugate of the network is actually computed. This is explicitly given in the references but is not found in the main text or appendix.
* For the additional amount of computation that is supposedly needed to perform the mirroring, does the CPMLA method outperform standard Langevin methods during test-time, i.e. with a fixed energy model? This should be addressed to accurately consider the practicality of this method.
* Could the authors clarify the motivation behind Eq. 8 and the surrounding exposition as ``practical application'', if the version of MLA used in Algorithm 1 comes from discretizing Eq. 7? Same with the seemingly contradictory paragraph above Algorithm 1. I believe it has to do with the noise distribution $\xi \sim \mathcal{N}(0, \nabla^2 G)$ but I am not sure.
* While theoretically motivated, the practical approximations are hidden away in the text. For example, diagonal Hessian approximation (l.263) is only mentioned once, but is critical to maintaining computational feasibility. The paper could use a paragraph explaining all the approximations made, perhaps somewhere next to Algorithm 1.
* There is no comparison towards classical MCMC methods when performing sampling from EBMs. However I am unfamiliar with the energy-based-model sampling literature, so perhaps this is not critical.
* Figure 2 appears to have an iteration cut-off at an arbitrarily chosen $T=21$, where it seems that CoopFlow may overtake CPMLA. In the following table, CPMLA is beaten by CoopFlow at $T=30$. While I appreciate that CoopFlow has significantly more parameters which may cause memory concerns, the iteration count is not enough to discern whether the proposed CPMLA is a practical method. There is no comparison towards methods with similar parameter count either.

[1] De Bortoli, V., Durmus, A., Pereyra, M., & Vidal, A. F. (2021). Efficient stochastic optimisation by unadjusted Langevin Monte Carlo: Application to maximum marginal likelihood and empirical Bayesian estimation. Statistics and Computing, 31, 1-18.

**Questions:**

* (Critical) Clarify that the convergence result in Section 5 does not apply to the training, but rather to sampling for a fixed $G$.
* The assumptions required for Section 5 are not complete: $G$ needs to be $\mathcal{C}^3$.
* Please define relaxed log-concavity.
* Please add a motivation as to where $\log \det$ comes from when training CP-Flow, see Huang et al. (2020).
* (Eq 2) The stated ``Langevin Monte Carlo'' is more commonly called the unadjusted Langevin algorithm (ULA). I suggest using ULA and citing the required references.
* How is $\nabla G^*$ computed? The authors initially claim that you can compute the conjugate using convex optimization, but this does not give you the gradient.
* (l.242) ``conugate''
* Please weaken the claim at the start of Section 5 about convergence of DNNs. This is a special case of general theory.
* (Appendix F) "our work introduces" -> "our work considers", mirror log-Sobolev has already been introduced in Jiang 2021.

---

> ### Author Response · Authors · 2024-11-24
> **Thank you for your comments and suggestions. (Block 1)**
>
> **Q1:** There is no mention as to how the inverse map is computed in practice.
> **A1:** The computation of the inverse of the mirror map, $(\nabla G)^{-1}$, is discussed in line 225 and further detailed in Appendix D. Specifically, $(\nabla G)^{-1}$ is computed as the convex conjugate of the mirror map $\nabla G$, as defined within the framework of our convex potential flow (CP-Flow). The convexity of CP-Flow ensures both the existence and uniqueness of this inverse. Our approach leverages conjugate convex optimization to efficiently compute $(\nabla G)^{-1}$, enabling CPMLA to smoothly transition between primal and dual spaces without introducing significant computational overhead.
>
>
> **Q2:** The theory claims to show convergence of the CPMLA algorithm during training, but does not consider the learnable parameters.
> **A2:** In our theoretical analysis, Assumptions 5.1 through 5.4 ensure that the convergence properties hold even as the parameters of CP-Flow and the EBM evolve during training. Importantly, the training parameters do not alter the underlying network structure or its properties. This is guaranteed by the structure of $ G $, as stated in Equation 5. Notably, the convex potential retains strong convexity with respect to the input $ x $, independent of the learnable parameters $ \vartheta $, ensuring the persistence of the unique inverse map, the retention of optimality via Brenier's theorem, and the continued ability to optimize the loss function without affecting convergence guarantees.
>
>
> **Q3:** Critical typos in Algorithm 1: $\nabla f$ is evaluated in dual space. And it should be $\nabla G(\nabla G^*y_t)$ in the sqrt.
>
> **A3:** These are not typos, indeed these are correct formulation of Mirror Langevin Algorithm. We provide a detailed explanation below.
>
> - **Gradient Descent in the Dual Space:** The method of evaluating the gradient in the dual space, as presented in Algorithm 1, follows the standard procedure for mirror gradient descent and mirror Langevin Dynamics. Specifically, previous works in this field, including those by Hsieh et al. (2018) and Jiang (2021), have consistently evaluated the gradient in the dual space. This approach aligns with the fundamental design of the mirror Langevin dynamics and is fully consistant with the established literature on this topic.
>
> - **Square Root in Equation 8:** We would like to clarify that the term $\nabla^2 G(y_t)$, which appears under the square root, is indeed correctly placed according to the derivations outlined in the reference by Jiang (2021), particularly in Section 4.3, which discusses $MLA_{\text{AFD}}$. In this context, $\nabla^2 G$ represents the Hessian of the convex potential and is properly evaluated at each iteration, reflecting the standard formulation for alternative forward discretization.
>
>
> **Q4:** Does CPMLA with additional computation outperform standard Langevin methods during test time with a fixed EBM?
> **A4:** CPMLA outperforms CoopFlow in terms of both inference efficiency and sample quality, and CoopFlow itself already provides significant improvements over standard Langevin methods. Therefore, CPMLA represents a meaningful advancement in comparison with the standard Langevin techniques. Standard Langevin sampling methods are computationally demanding and non-mixing. The additional computation introduced by CPMLA is designed to address these limitations and create a better sampling algorithm. Specifically, CPMLA demonstrates a clear performance advantage over standard Langevin Monte Carlo methods under similar test conditions, particularly when using a fixed EBM on CIFAR-10, as shown in Appendix I, Figure 6.
>
>
> **Q5:** What is the motivation behind Equation 8? Algorithm 1 seems contradictory with Equation 7 and 8.
>
> **A5:**
> - **Motivation behind Equation 8:**
>    Equation 8 and Algorithm 1 use a discretization method where the gradient descent and SDE steps are separated. This design is deliberate, as it allows the SDE step’s drift term to become zero, simplifying the implementation and improving accuracy when discretizing. By removing the drift term in the SDE, we achieve a straightforward numerical scheme for the noise term that is computationally efficient. Also the Hessian of the convex potential plays a key role in adapting the sampling steps to the underlying geometry of the data manifold. The motivation for this inclusion is well-documented in the literature, particularly in the work by Jiang (2021), which discusses the alternative forward discretization scheme and highlights its advantages in terms of convergence and bias reduction.
>
> - **Clarification on Apparent Contradiction:**
>    The points raised in lines 259 and 269 of the paper clearly explain how Algorithm 1 and Equation 8, though differing slightly in form, represent the same underlying discretization approach for MLA. It IS related to the noise distribution $\xi \sim \mathcal{N}(0, \nabla^2 G)$.

---

> ### Author Response · Authors · 2024-11-24
> **Thank you for your comments and suggestions. (Block 2)**
>
> **Q6:** The practical approximations are hidden away in the text, especially the diagonal Hessian approximation.
>
> **A6:** The diagonal Hessian approximation is a practical approximation employed in our method mentioned on line 263. This simplification allows us to avoid the full Hessian calculation, significantly reducing computational costs without compromising sampling accuracy. This choice is particularly advantageous in high-dimensional settings where full Hessian computations are impractical. We will revise the manuscript to include a paragraph of practical modification near Algorithm 1.
>
>
> **Q7:** There is an arbitrarily chosen cut-off $T=21$ in Figure 2. There is no comparison with similar parameter count.
>
> **A7:**
>
> - **Clarification on Iteration Cut-Off:** The iteration cut-off in Figure 2 was selected to showcase CPMLA's rapid convergence and inference efficiency compared to CoopFlow. We acknowledge that it may appear that CoopFlow could overtake CPMLA if allowed more iterations. However, our goal was to emphasize the significant efficiency gains CPMLA provides early on. Specifically, as shown in Table 2, CPMLAprt achieves a FID score of 21.43 at $ T = 20 $, which is comparable to CoopFlow's FID of 21.16 at $ T = 30 $. This represents a 33% reduction in the number of iterations required to reach similar performance, demonstrating CPMLA's practical advantages in scenarios where faster sampling is critical.
>
> - **Memory Efficiency and Parameter Count:** The key strength of CPMLA lies in achieving better sampling efficiency with fewer parameters, rather than generating higher-quality images with the same parameter count. The significant difference between 0.26M and 28.78M parameters highlights CPMLA’s efficiency, which is particularly important for applications with limited computational resources. While our primary comparisons focused on CoopFlow, we are open to extending future analyses to include methods with comparable parameter counts to provide a more comprehensive evaluation.
>
>
> **Q8:** Please define relaxed log-concavity.
>
> **A8:** Relaxed log-concavity is a key assumption of target distributions as we have mentioned in Line 75 and 109. It is a less restrictive version of log-concavity, permitting some flexibility while still maintaining key properties like unimodality or a tendency towards a peaked structure in the function. To begin, we first define **strongly log-concave**. Just as a strongly convex function ensures well-conditioned geometry for fast, unbiased optimization, a strongly log-concave distribution ensures a similar well-conditioned structure, facilitating efficient sampling of true samples. This property is explored in several works, such as [1][2][3][4][5].
>
> A function $ f(x) $ is strongly log-concave if the logarithm of the function is strongly concave, i.e., for any two points $ x_1 $ and $ x_2 $ and for any $ \lambda \in [0, 1] $,
>
> $$
> f(\lambda x_1 + (1 - \lambda) x_2) \geq f(x_1)^\lambda f(x_2)^{1 - \lambda}.
> $$
> In our work, **relaxed log-concavity** is a broader condition compared to "strongly log-concave" distributions. Specifically, a distribution satisfying a Logarithmic Sobolev Inequality (LSI) can be considered an example of a "relaxed log-concave" distribution. The LSI condition ensures certain concentration properties similar to those of log-concave distributions, even though the global curvature constraints are less stringent.
>
> [1] Alain Durmus and Eric Moulines. Non-asymptotic convergence analysis for the unadjusted langevin algorithm. Annals of Applied Probability,Annals of Applied Probability, Dec 2016a.
>
> [2] Alain Durmus and Eric Moulines. High-dimensional bayesian inference via the unadjusted langevin algorithm. Bernoulli,Bernoulli, Dec 2016b.
>
> [3] Xiang Cheng, Niladri S. Chatterji, PeterL. Bartlett, and MichaelI. Jordan. Underdamped langevin mcmc: A non-asymptotic analysis. Cornell University - arXiv,Cornell University - arXiv, Jul 2017.
>
> [4] Arnak S. Dalalyan and Avetik Karagulyan. User-friendly guarantees for the langevin monte carlo with inaccurate gradient. Stochastic Processes and their Applications, 129(12):5278–5311, Dec 2019. doi: 10.1016/j.spa.2019.02.016. URL http://dx.doi.org/10.1016/j.spa.2019.02.016.
>
> [5] Wenlong Mou, Yuanlin Ma, MartinJ. Wainwright, PeterL. Bartlett, and MichaelI. Jordan. High- order langevin diffusion yields an accelerated mcmc algorithm. arXiv: Machine Learning,arXiv: Machine Learning, Aug 2019.

---

> > ### Comment · Reviewer_67JH · 2024-11-24
> > **Response**
> >
> > I thank the authors for their detailed responses to my concerns and considered changes to the manuscript. I still have questions in order to fairly assess the proposed method.
> >
> > I am happy with the response to point 6.
> >
> > **Q1.** Please clarify directly how the inner optimization problem in Appendix D is solved. Inner optimization loops do not seem like an , and this takes up a significant portion of the CP-Flow paper. It is a-priori not clear how backpropagation will work for training such Langevin maps. The presence of many apparent inner loops (such as the minimization for (6), as well as presumed Monte Carlo sampling w.r.t Rademacher $v$) makes this concerning.
> >
> > **Q2.** While the assumptions may still be satisfied during the training, the target distribution $\pi \propto p_\theta(x)$ (given exactly by your energy based model) is changing, and Theorem 5.5 in its current form does not apply when updating $\theta$. Please address as suggested in my initial comment.
> >
> > **Q3.** Apologies, there was a typo in my correction, it should read $\nabla^2 G(\nabla G^* y_t)$. I do not have a problem with inverting the Hessian of the convex conjugate, but rather the points at which gradients are evaluated. The correct identity is
> > $D^2 h(x) \cdot D^2 h^*(Dh(x)) = I$ (even correctly stated in Lemma E.1). The notation in the mirror descent/Langevin literature is generally misleading, so let me explain my reasoning as to why I believe that the current formulation is incorrect. I will assume that $x_k$ and $X_t$ lie in the primal space $\mathcal{X} = \mathbb{R}^d$, and $y_k, Y_t$ lie in the dual space $\mathcal{X}^* = (\mathbb{R}^d)^*$. Then, we have $\nabla G: \mathcal{X} \rightarrow \mathcal{X}^*$, $\nabla G^*: \mathcal{X}^* \rightarrow \mathcal{X}$, and $f : \mathcal{X} \rightarrow \mathbb{R}$. This immediately raises questions about why $\nabla f$ is evaluated at a dual point $y \in \mathcal{X}^*$.
> >
> > $\textrm{MLA}_{\textrm{AFD}}$ in (Jiang 2021) then states in their (12), replacing their instances of $\phi$ with your $G$, to solve
> >
> > $dy_t = \sqrt{2 [\nabla^2 G^* (y_t)]^{-1}} dW_t$ for $y_0 = \nabla G(x_{k+1/2})$.
> >
> > Applying the aforementioned equality with $h=G^*$, we get that $[\nabla^2 G^* (y_t)]^{-1} = \nabla^2 G (\nabla G^* y_t)$, where $(\nabla G^* y_t)$ lives in the primal space $\mathcal{X}$. Algorithm 1 will make sense with Equation 8 if
> >
> > 1. In Equation (8), $\nabla^2 G(y_t)$ is replaced by $\nabla^2 G(\nabla G^* y_t)$,
> >
> > 2. On line 284, $\nabla f(\hat{y}_{i,k})$ is replaced with $\nabla f(\nabla G^* \hat{y})$.
> >
> > I hope the authors can resolve this. Perhaps additionally consider using $f_\theta$ instead of $f$ in the statement of Alg. 1.
> >
> > **Q4.** I did not understand Appendix I on first reading. I am happy with the response.
> >
> > **Q5.** I think it makes more sense after re-reading the equations, thank you. I strongly suggest rewriting the paragraph that starts on line 269. My understanding is that it says the formulation in Algorithm 1 avoids re-calculating $\nabla G(x_k)$ in the first line of Equation (8) since the dual is already available, and can be simply written a computational tool to avoid extraneous mapping between the primal and dual space.
> >
> > **Q6.** I am happy with the response.
> >
> > **Q7.** I strongly suggest the authors to carefully address point 7 by adding wall clock time comparison and extension to further iterations to improve the presentation, as noted additionally by Reviewer siTL. As acknowledged by the author, the figure does indeed suggest different behavior at T=30. However, it is misleading to directly give claims of "superior inference speed" on l.403 with incomplete information, and thus higher values of T should still be added into Figure 2. I suggest also to slightly rewrite the aforementioned paragraph.
> >
> > **Q8.** The definition given in the authors' response is of standard log-concavity, not strong log-concavity. I would suggest directly using LSI as the distributional assumption, or clarifying the exact definition of relaxed log-concavity early in the paper. If the presumed definition is given by Assumptions 5.1-5.4, then perhaps it can be reformulated as "standard regularity assumptions (Jiang, 2021)".
> >
> > Typo(?), $\mathcal{K}$ in line 224: $G^*(y) = \max_{x \in \mathcal{K}} ...$
> >
> > **Summary:** I am still not convinced by the theoretical components and experimental methodology, where I believe there are instances of $\nabla G^*$ missing, as explained above. On the experimental front, I appreciate the authors inclusion of non-mixing of standard Langevin methods for EBMs. However the comparison with the closest work Coopflow (Xie et al. 2022b) is not sufficiently evaluated, as iteration count is not everything, especially with the (presumed) requirement of an inner optimization loop to compute the convex conjugate.

---

> > > ### Author Response · Authors · 2024-11-25
> > > **Thanks you for your comments and suggestions. (Block 1)**
> > >
> > > **Q1:** Please clarify directly how the inner optimization problem of CP-Flow in Appendix D is solved.
> > >
> > > **A1:** We would like to clarify that the details of solving the inner optimization problem, including the minimization in Equation 6 and the Monte Carlo sampling with respect to Rademacher variables, are discussed in depth in Sections 3.1 and 3.2 in [1].
> > >
> > > Regarding the computation of the inverse of CP-Flow (Algorithm 1 in [1]), we transform the problem into computing the convex conjugate. Since the underlying optimization problem is convex, we solve it using the L-BFGS algorithm.
> > >
> > > As for estimating the log probability (Algorithm 2 in [1]), directly backpropagating through the log determinant estimator is not ideal due to significant memory overhead. Instead, this can be converted into a convex optimization problem. As outlined in Appendix C, specifically in Equation 16, the problem is reformulated to compute $ v^{\top} H^{-1} $. This can be achieved by solving a quadratic optimization problem, which we solve efficiently using the conjugate gradient method. This approach enables an iterative solution without explicitly inverting $ H $, thus reducing memory usage.
> > >
> > > We will include additional details in the Appendix to clarify these points further.
> > >
> > > [1] Chin-Wei Huang, Ricky T. Q. Chen, Christos Tsirigotis, and Aaron Courville. Convex Potential Flows: Universal Probability Distributions with Optimal Transport and Convex Optimization. Learning, 2020.
> > >
> > >
> > > **Q2:** The target distribution $\pi \propto p_{\theta}(x)$ is changing during the training process.
> > >
> > > **A2:** We would like to clarify that the target distribution $\pi$ is defined as proportional to the data distribution $p_{\text{data}}(x)$, which does not change during training.
> > >
> > > **Q3:** Typos in Algorithm 1.
> > >
> > > **A3:** Thank you for your detailed explanation and for pointing out the typo. We acknowledge that $G$ is defined in the primal space, so the argument should indeed be $x$. We have correctly written this in Equation 7, Lemma E1, and other parts of the paper. Similarly, $f$ is also defined in the primal space. We have correctly written this in Equations 1, 3, 8, and elsewhere. We will revise the manuscript to reflect your correction, ensuring that the notation is consistent and the formulation clearer.
> > >
> > > **Q5:** Rewriting the paragraph that starts on Line 269.
> > >
> > > **A5:** Thank you for your thoughtful comment and for re-reading the equations. We will rewrite the paragraph starting at line 269 to more clearly express that the formulation in Algorithm 1 avoids recalculating $ \nabla^2 G $ in the first line of Equation 8 because the dual is already available.
> > >
> > >
> > > **Q8:** I would suggest directly using LSI as the distributional assumption, or clarifying the exact definition of relaxed log-concavity early in the paper.
> > >
> > > **A8:** Thanks for your suggestions. In the revised version, we will clarify the concept of relaxed log-concavity and explicitly assume that the target distribution satisfies the LSI condition.

---

> > > ### Author Response · Authors · 2024-11-25
> > > **Thank you for your comments and suggestions. (Block 2)**
> > >
> > > **Q7:** I strongly suggest the authors carefully address point 7 by adding a wall clock time comparison and extending the analysis to further iterations to improve the presentation.
> > > **A7:** Thank you for the suggestion. Below, we summarize the additional experiments we have included:
> > >
> > > **Table 1:** A comparison of wall clock time for inferring 1,000 images of similar quality on the CIFAR-10 dataset
> > > |                    | CPMLAprt (T = 20) | CoopFlow (T = 30) |
> > > |:------------------:|:-----------------:|:-----------------:|
> > > | Time (s/1k images) | 16.14             | 16.84             |
> > >
> > > As shown in Table 1, our model, CPMLAprt, outperforms CoopFlow in terms of inference speed.
> > >
> > > Table 1 compares the wall clock time required to generate 1,000 images of similar quality (with FID scores of 21.43 vs. 21.16) on the CIFAR-10 dataset for CPMLAprt (T = 20) and CoopFlow (T = 30). To account for potential initialization and other sources of error, we calculated the average wall clock time by generating 50,000 images when computing the FID score. The results show that, although CPMLA involves additional computations, the lower iteration count and fewer parameters more than compensate for this, leading to a superior inference speed compared to the baseline.
> > >
> > > **Table 2:** A comparison of FID scores under different iteration counts
> > > |                   | 10    | 20    | 24    | 27    | 30    |
> > > |:-----------------:|:-----:|:-----:|:-----:|:-----:|:-----:|
> > > | CoopFlow (T = 30) | 57.26 | 30.74 | 25.03 | 22.45 | 21.16 |
> > > | CPMLA (T = 20)    | 56.40 | 26.92 | 24.76 | 23.47 | 22.35 |
> > >
> > > As shown in Table 2, the FID score of CPMLA is lower than that of CoopFlow when the iteration count is 20, and is comparable when the iteration count reaches 30. We would like to emphasize that the key strength of CPMLA lies in achieving better sampling efficiency with fewer parameters. When comparing the difference in FID scores relative to the differences in parameter counts, the former is minimal, as shown in the following Table 3.
> > >
> > > **Table 3:** A comparison of FID scores and the number of parameters across different EBMs and flow-based frameworks
> > > | Methods       | Number of Parameters | FID   |
> > > |:-------------:|:--------------------:|:-----:|
> > > | NT-EBM        | 23.8M                | 78.12 |
> > > | GLOW          | 44.2M                | 45.99 |
> > > | EBM_FCE       | 44.9M                | 37.30 |
> > > | CoopFlow      | 45.9M                | 21.16 |
> > > | Flow++ only   | 28.8M                | 92.10 |
> > > | VAEBM         | 135.1M               | 12.16 |
> > > | **CPMLA**     | **17.39M**           | **22.35** |
> > >
> > > Table 3 demonstrates that parameter count of CPMLA is significantly fewer than other methods, while still maintaining high image quality. This highlights the efficiency of CPMLA in terms of both parameter usage and performance.

---

> > > > ### Comment · Reviewer_67JH · 2024-11-25
> > > >
> > > > I thank the authors for their fast and detailed response regarding my latest set of concerns. I am now happy with the experimental side, and the additional details that will be added to the appendix.
> > > >
> > > > Regarding the theoretical component, I have the same concerns as before about Theorem 5.5. The authors state that $\pi \propto \exp(-f)$ is actually the data distribution, but how is this (or $f$) defined? I understand that MLA, in particular iterating the Langevin step $y - \eta \nabla f(\nabla G^* y) + \sqrt{2\eta} \xi$, will converge to a biased version of the stationary distribution $\exp(-f)$ after passing to primal space. However, my understanding of the co-operative learning is that $f$ in the gradient step is actually the EBM being jointly optimized. This makes the most sense to me as $f$ is intractable for the image experiments suggested.
> > > >
> > > > I am willing to raise my score if the theoretical component is correct or corrected.

---

> ### Author Response · Authors · 2024-12-01
>
> Thank you for your comment.
>
> The Langevin step in Algorithm 1 should be $y - \eta \nabla f_{\theta}(\nabla G^*y) + \sqrt{2\eta}\xi$. To address the gap introduced by this adjustment, we have revised Theorem 5.5 accordingly.
>
> We prove that the total variation distance satisfies $d_{TV}(\rho_t, p_{\text{data}}) < \delta$, where $\delta$ is a small constant. A sketch of the proof is provided below.
>
> We decompose the total variation distance between $\rho_t$ and $p_{\text{data}}$ into three terms, providing controls for each. This revised formulation guarantees that $\rho_t$ converges to the data distribution, thereby addressing the reviewer's concerns.
>
> Specifically, let $q_{\vartheta^*}$ denote the target distribution of CP-Flow and $p_{\theta^*}$ the target of the EBM. For the total variation distance $d_{TV}(\rho_t, p_{\text{data}})$, we decompose it into three terms:
> $ d_{TV}(\rho_t, p_{\text{data}}) \leq d_{TV}(\rho_t, p_{\theta^*}) + d_{TV}(p_{\theta^*}, q_{\vartheta^*}) + d_{TV}(q_{\vartheta^*}, p_{\text{data}}). $
>
> - The first term is controlled under the original Theorem 5.5 (substituting $\delta$ with $\delta_1$). Using Pinsker's inequality we establish:
>   $
>   d_{TV}(\rho_t, p_{\theta^*}) \leq \sqrt{\frac{1}{2} D_{KL}(\rho_t \| p_{\theta^*})} < \sqrt{\frac{\delta_1}{2}}.
>   $
> - The second term can be controlled because Langevin MCMC only runs for $T$ steps rather than converging to its stationary distribution. By analyzing the Fokker-Planck equation of Langevin sampling:
>   $
>   \frac{\partial p_t(x)}{\partial t} = -\nabla_x \cdot \left(p_t(x) \frac{\eta^2}{2} \nabla_x f_\theta(x)\right) + \frac{\eta^2}{2} \nabla_x^2 p_t(x),
>   $
>   we estimate the incremental change as $d_{TV}(p_t, p_{t-1}) \leq \sqrt{\frac{1}{2} D_{KL}(p_t \| p_{t-1})} \sim O(\eta)$. Summing over $T$ steps gives:
>   $d_{TV}(p_{\theta^*}, q_{\vartheta^*}) \sim O(\eta \sqrt{T}). $
>   Thus, the second term can be bounded by $\delta_2$ by balancing $\eta$ and $T$, particularly in the later stages of training.
> - The third term leverages the universality property of CP-Flow (Theorem 3 in [11]). Given that the initial noise distribution is absolutely continuous with respect to the Lebesgue measure, there exists a sequence $q_{\vartheta_n}$ such that
>   $d_{TV}(q_{\vartheta_n}, p_{\text{data}}) < \delta_3 \text{ as } n > N. $The optimality of CP-Flow (Theorem 4 in [11]) further guarantees almost sure convergence in distribution of $q_{\vartheta_n}$ to the optimal Brenier map $q_{\vartheta^*}$, ensuring that $d_{TV}(q_{\vartheta^*}, p_{\text{data}}) < \delta_3$.
>
> Combining these results, we conclude that $d_{TV}(\rho_t, p_{\text{data}}) < \delta = \sqrt{\frac{\delta_1}{2}} + \delta_2 + \delta_3$. While this conclusion is weaker than our original claim, it remains sufficient to establish the convergence of Algorithm 1.
>
> We will revise the theorem with detailed proof.

---

### Official Review · Reviewer_siTL · 2024-11-03

**Soundness:** 3
**Presentation:** 3
**Contribution:** 2
**Rating:** 3
**Confidence:** 3

**Summary:**

This paper expects to enhance sampling quality and inference efficiency by introducing the convex potential flow (CP-Flow) to energy-based models (EBM). Different from standard EBM, this paper not only approximates the gradient of log density of target distribution but also considers the SDE path toward the target. The benefits of enforcing the SDE path to optimal transport in this paper lie in decreasing the mixing time of the convergence expected for the continuous version.

**Strengths:**

1. The motivation for introducing the SDE path requirement to EBM to enhance the sampling efficiency is interesting and reasonable from an intuition perspective.
2. This paper is well written; the intuition, implementation, and experiments are clear.

**Weaknesses:**

1. The novelty of this paper is not enough; controlling the path of generation with additional models beyond EBM is not novel and is highly similar to the baseline Coopflow. This paper only replaces the normalizing flow (in Coopflow) with mirror Langevin Dynamics.
2. The analysis of this paper does not match their claim:
    1. Theorem 5.5 is only established when the approximation error of EBM and CP-Flow are both 0. That means this analysis does not consider how the approximation error affects the final inference efficiency. Since this paper considers a generative model rather than a sampling algorithm, I think the approximation error is necessary to be considered, just as shown in [Chen2022].
    2. The analysis of Theorem 5.5 does not include any novel technique, which just utilizes the framework proposed in [Vempala2019]  to analyze a sampling algorithm. Moreover, the Fokker-Planck equation is actually given by [Jiang 2021]
    3. Line. 37—Line. 38, authors argue that MCMC tends to get trapped in local modes and has a slow convergence. Actually, the analysis of this paper shows that CPMLA achieves KL convergence with the same gradient complexity as that in unadjusted Langevin Algorithm [Vempala2019] w.r.t. the dimension $d$ and error tolerance $\epsilon$ dependence. Besides, it is worse than underdamped Langevin Dynamics [Zhang2023].
3. The experiment does not show a significant advantage compared with Coopflow. I suggest that the authors provide an FID curve when T is larger than 20. Since the decrease in the tendency of Coopflow is sharper than that of CPMLA, in Table 2, I suggest that the authors also show the performance of CPMLA when T=30.

I am willing to raise my score if the authors solve my concerns.



[Chen2022] Chen S, Chewi S, Li J, et al. Sampling is as easy as learning the score: theory for diffusion models with minimal data assumptions[J]. arXiv preprint arXiv:2209.11215, 2022.

[Jiang2021] Qijia Jiang. Mirror langevin monte carlo: the case under isoperimetry. Neural Information Processing Systems, 2021.

[Vempala2019] Vempala S, Wibisono A. Rapid convergence of the unadjusted langevin algorithm: Isoperimetry suffices[J]. Advances in neural information processing systems, 2019, 32.

[Zhang2023] Zhang S, Chewi S, Li M, et al. Improved discretization analysis for underdamped Langevin Monte Carlo[C]//The Thirty Sixth Annual Conference on Learning Theory. PMLR, 2023: 36-71.

**Questions:**

Please check the weakness part.

---

> ### Author Response · Authors · 2024-11-24
> **Thank you for your comments and suggestions. (Block 1)**
>
> We sincerely thank the reviewer for their thoughtful and constructive feedback. We will address reviewer's questions as below.
>
> **Q1:** The novelty of this paper is not enough; controlling the path of generation with additional models beyond EBM is not novel and is highly similar to the baseline CoopFlow. This paper only replaces the normalizing flow (in CoopFlow) with mirror Langevin Dynamics.
> **A1:** We respectfully disagree with the assessment that our work “only replaces the normalizing flow (in CoopFlow) with mirror Langevin Dynamics.” CPMLA is the first to introduce the Mirror Langevin Algorithm (MLA) as a sampling technique within deep neural networks, whereas previous MLAs lacked large-scale experimental validations and only used predefined fixed mirror maps. Compared to CoopFlow, CPMLA is a second-order sampling algorithm that utilizes the Hessian of CP-Flow to better capture the underlying geometry of the data manifold, thereby achieving more efficient sampling. Experimental results demonstrate that CPMLA outperforms CoopFlow in terms of inference efficiency and FID scores, while using only 0.9% of CoopFlow’s parameter count. This highlights the effectiveness of our dynamic mirror map and the practical advantages of CPMLA in terms of computational efficiency and model complexity.
>
> In summary, CPMLA introduces a different approach to sampling in EBMs through a dynamic, geometry-aware mirror map. We believe these innovations clearly differentiate CPMLA from CoopFlow.
>
>
> **Q2:** The approximation error is necessary to be considered.
> **A2:** In line 401 and 456, empirically, our excellent FID scores demonstrate that CPMLA provides a reliable approximation of the target distribution, without introducing bias.
>
>
> **Q3:** The analysis of Theorem 5.5 does not include any novel technique compared with [Vempala 2019] and [Jiang 2021].
> **A3:** Vempala's convergence framework is the analysis of the Unadjusted Langevin Algorithm (ULA), whereas we focus on the analysis of the Mirror Langevin Algorithm (MLA). In contrast to Vempala's framework, which uses the standard norm $||z||$, we introduce second-order information by incorporating a geometry-aware mirror flow. Specifically, we define the extended norm as $ ||z||_{\nabla^2 G}^2 = z^{\top} {\nabla^2 G} z $ in the dual space. This specific adaptation is crucial for analyzing the dynamics in our setup, as CPMLA utilizes a dynamically adjusted mirror map within the CP-Flow structure.
>
> In contrast to Jiang's work, our approach leverages the bounded eigenvalues of the CP-Flow Hessian to enhance theoretical guarantees. This enables us to exert more precise control over the behavior of the dynamics in dual space, ensuring more robust convergence, particularly in high-dimensional settings. Notably, both Vempala's and Jiang's work are foundational and widely recognized in the field for analyzing the convergence of sampling algorithms, and in fact, Jiang’s analysis of continuous Mirror Langevin Dynamics (MLD) is itself inspired by Vempala’s methods.

---

> > ### Author Response · Authors · 2024-11-24
> > **Thank you for your comments and suggestions. (Block 2)**
> >
> > **Q4:** The gradient complexity of CPMLA is the same as Unjusted Langevin Algorithms (ULA) and worse than Underdamped Langevin Dynamics (ULD).
> >
> > **A4:** We would like to clarify our improvements of CPMLA in 4 folds:
> > - **Assumption Relaxation**: Unlike ULA, which requires a strongly log-concave target distribution to ensure convergence, CPMLA relaxes this assumption. Our method guarantees convergence even under a *relaxed log-concave condition*. This extension is crucial, as it allows CPMLA to perform well on more complex, non-log-concave distributions, which are often encountered in real-world high-dimensional data scenarios.
> >
> > - **Improved Mixing Time**: CPMLA also has distinct advantages in mixing time. Traditional methods such as ULA, hit-and-run, and ball walk exhibit mixing times that scale with the square of the condition number of the space, $\mathcal{C}_{\mathcal{K}}^2$. This scaling can severely impact gradient complexity in ill-conditioned domains. In contrast, CPMLA’s mixing time is independent of the condition number, which enables it to achieve consistent gradient complexity even in poorly conditioned spaces. This property is a key factor behind CPMLA’s robustness and efficiency across diverse sampling tasks, particularly where traditional methods may struggle due to condition number dependence.
> >
> > - **Comparison with Underdamped Langevin Dynamics**: CPMLA differs from ULD in its approach to geometry adaptation. By introducing the mirror map in a dual space, CPMLA optimizes sampling efficiency on data manifolds that may not be well-suited to Euclidean sampling dynamics. This geometry-aware adaptation is particularly advantageous in applications involving high-dimensional or structured data, where ULD may not fully leverage such manifold-specific information. Our empirical results reflect these advantages, showing that CPMLA achieves superior performance on complex data distributions compared to both ULA and ULD.
> >
> > - **Empirical Benefits of CPMLA**: The practical advantages of CPMLA are apparent in our experiments. The dynamic mirror map allows CPMLA to achieve faster practical convergence and better sample quality on real-world high-dimensional tasks, addressing the mode-trapping issue that often impacts MCMC methods on non-convex landscapes. This balance between theoretical rigor and practical effectiveness is a key advantage of CPMLA.
> >
> > We hope this clarifies the advantage and uniqueness of CPMLA compared with ULA and ULD.
> >
> >
> > **Q5:** The experiment does not show a significant advantage compared with CoopFlow.
> >
> > **A5:** We would like to clarify that the advantages of CPMLA on inference efficiency and FID score over CoopFlow are already demonstrated in Figure 2 and Table 1 of our paper. CPMLA exhibits a more rapid decrease in FID scores with the same number of sampling steps and achieves better FID scores by the end of the sampling process. While CoopFlow may achieve slightly better FID scores at $T=30$, this comes at the cost of a significantly larger parameter count. Specifically, as shown in Table 1, CPMLA operates with only 0.9% of the parameters required by CoopFlow’s normalizing flow component (0.27M vs. 28.78M), illustrating CPMLA’s efficiency. Despite this vast difference in parameters, CPMLA’s FID scores remain close to CoopFlow’s, which highlights CPMLA's effective sampling quality with minimal computational overhead.

---

> ### Author Response · Authors · 2024-11-25
> **Thank you for your comments and suggestions. (Block 3)**
>
> Below, we summarize the additional experiments we have included: a comparison of wall clock time, a comparison of FID scores in further iterations and a comparison of FID scores and the number of parameters across different EBMs and flow-based frameworks.
>
> **Table 1:** A comparison of wall clock time for inferring 1,000 images of similar quality on the CIFAR-10 dataset
> |                    | CPMLAprt (T = 20) | CoopFlow (T = 30) |
> |:------------------:|:-----------------:|:-----------------:|
> | Time (s/1k images) | 16.14             | 16.84             |
>
> As shown in Table 1, our model, CPMLAprt, outperforms CoopFlow in terms of inference speed.
>
> Table 1 compares the wall clock time required to generate 1,000 images of similar quality (with FID scores of 21.43 vs. 21.16) on the CIFAR-10 dataset for CPMLAprt (T = 20) and CoopFlow (T = 30). To account for potential initialization and other sources of error, we calculated the average wall clock time by generating 50,000 images when computing the FID score. The results show that, although CPMLA involves additional computations, the lower iteration count and fewer parameters more than compensate for this, leading to a superior inference speed compared to the baseline.
>
> **Table 2:** A comparison of FID scores under different iteration counts
> |                   | 10    | 20    | 24    | 27    | 30    |
> |:-----------------:|:-----:|:-----:|:-----:|:-----:|:-----:|
> | CoopFlow (T = 30) | 57.26 | 30.74 | 25.03 | 22.45 | 21.16 |
> | CPMLA (T = 20)    | 56.40 | 26.92 | 24.76 | 23.47 | 22.35 |
>
> As shown in Table 2, the FID score of CPMLA is lower than that of CoopFlow when the iteration count is 20, and is comparable when the iteration count reaches 30. We would like to emphasize that the key strength of CPMLA lies in achieving better sampling efficiency with fewer parameters. When comparing the difference in FID scores relative to the differences in parameter counts, the former is minimal, as shown in the following Table 3.
>
> **Table 3:** A comparison of FID scores and the number of parameters across different EBMs and flow-based frameworks
> | Methods       | Number of Parameters | FID   |
> |:-------------:|:--------------------:|:-----:|
> | NT-EBM        | 23.8M                | 78.12 |
> | GLOW          | 44.2M                | 45.99 |
> | EBM_FCE       | 44.9M                | 37.30 |
> | CoopFlow      | 45.9M                | 21.16 |
> | Flow++ only   | 28.8M                | 92.10 |
> | VAEBM         | 135.1M               | 12.16 |
> | **CPMLA**     | **17.39M**           | **22.35** |
>
> Table 3 demonstrates that parameter count of CPMLA is significantly fewer than other methods, while still maintaining high image quality. This highlights the efficiency of CPMLA in terms of both parameter usage and performance.

---

> > ### Comment · Reviewer_siTL · 2024-11-26
> >
> > I thank the authors for their detailed response and appreciate the additional experimental results.
> >
> > However, some of my concerns have not been solved.
> >
> > 1. The approximation error of EBM and CP-Flow should be clarified. Otherwise, we do not know how the training error (including the score approximation error and flow error) will affect the complexity of the inference process. The authors responded to this point with some experimental results, which can hardly be acceptable since it is a totally theoretical problem. If the authors claim that CPMLA provides a reliable approximation of the target distribution "without introducing bias," I hope they can provide some evidence, such as the approximation in some synthetic data.
> >
> > 2. In the rebuttal, the authors claim they relaxed the assumptions and said ULA requires the target distribution to satisfy strong log concave. Actually,  Theorem 1 of [Vempala2019] directly shows the convergence of ULA only requires log-Sobolev inequality (LSI), which is comparable with Assumption 5.1 in this paper. Moreover, since the LSI required in ULA is directly defined with the L_2 norm rather than the matrix norm shown in this paper, it can be verified more easily.
> >
> > 3. In the rebuttal, the authors claim they improved the mixing time independent of the condition number. Actually, their results depended on $\gamma^2/\beta^2$. Suppose $\gamma$ is similar to the smooth target distribution required in [Vempala2019], $\gamma^2/\beta^2$ will be the same as the smoothness, LSI constant dependence shown in Theorem 1 of [Vempala2019]. Therefore, I am not convinced that the mixing time is improved.
> >
> > Considering the above three points, I will maintain my score currently. I am willing to raise my score if the authors solve my concerns.

---

> ### Author Response · Authors · 2024-12-01
>
> We appreciate the reviewer's comments and would like to clarify the following points:
>
> **Q1:** The approximation error of EBM and CP-Flow should be clarified. I hope they can provide some evidence, such as the approximation in some synthetic data.
>
> **A1:** We have included empirical evidence in Figure 1 with synthetic data. CPMLA accurately estimates the eight-Gaussian mixture distribution density, demonstrating both approximation quality.
>
> We have revised Theorem 5.5 and proven that the total variation distance $d_{TV}(\rho_t, p_{\text{data}}) < \delta$ with a small constant $\delta$. We provide the proof sketch as follows.
>
> We decompose the total variation distance between $\rho_t$ and $p_{\text{data}}$ into three terms, providing controls for each. This revised formulation guarantees that $\rho_t$ converges to the data distribution, thereby addressing the reviewer's concerns.
>
> Specifically, let $q_{\vartheta^*}$ denote the target distribution of CP-Flow and $p_{\theta^*}$ the target of the EBM. For the total variation distance $d_{TV}(\rho_t, p_{\text{data}})$, we decompose it into three terms:
> $ d_{TV}(\rho_t, p_{\text{data}}) \leq d_{TV}(\rho_t, p_{\theta^*}) + d_{TV}(p_{\theta^*}, q_{\vartheta^*}) + d_{TV}(q_{\vartheta^*}, p_{\text{data}}). $
>
> - The first term is controlled under the original Theorem 5.5 (substituting $\delta$ with $\delta_1$). Using Pinsker's inequality we establish:
>   $
>   d_{TV}(\rho_t, p_{\theta^*}) \leq \sqrt{\frac{1}{2} D_{KL}(\rho_t \| p_{\theta^*})} < \sqrt{\frac{\delta_1}{2}}.
>   $
> - The second term can be controlled because Langevin MCMC only runs for $T$ steps rather than converging to its stationary distribution. By analyzing the Fokker-Planck equation of Langevin sampling:
>   $
>   \frac{\partial p_t(x)}{\partial t} = -\nabla_x \cdot \left(p_t(x) \frac{\eta^2}{2} \nabla_x f_\theta(x)\right) + \frac{\eta^2}{2} \nabla_x^2 p_t(x),
>   $
>   we estimate the incremental change as $d_{TV}(p_t, p_{t-1}) \leq \sqrt{\frac{1}{2} D_{KL}(p_t \| p_{t-1})} \sim O(\eta)$. Summing over $T$ steps gives:
>   $d_{TV}(p_{\theta^*}, q_{\vartheta^*}) \sim O(\eta \sqrt{T}). $
>   Thus, the second term can be bounded by $\delta_2$ by balancing $\eta$ and $T$, particularly in the later stages of training.
> - The third term leverages the universality property of CP-Flow (Theorem 3 in [11]). Given that the initial noise distribution is absolutely continuous with respect to the Lebesgue measure, there exists a sequence $q_{\vartheta_n}$ such that
>   $d_{TV}(q_{\vartheta_n}, p_{\text{data}}) < \delta_3 \text{ as } n > N. $The optimality of CP-Flow (Theorem 4 in [11]) further guarantees almost sure convergence in distribution of $q_{\vartheta_n}$ to the optimal Brenier map $q_{\vartheta^*}$, ensuring that $d_{TV}(q_{\vartheta^*}, p_{\text{data}}) < \delta_3$.
>
> Combining these results, we conclude that $d_{TV}(\rho_t, p_{\text{data}}) < \delta = \sqrt{\frac{\delta_1}{2}} + \delta_2 + \delta_3$. While this conclusion is weaker than our original claim, it remains sufficient to establish the convergence of Algorithm 1.
>
> We will revise the theorem with detailed proof.
>
>
> **Q2 & Q3:** ULA uses comparable assumptions with Assumption 5.1 in Theorem 1 of [Vempala2019]. I am not convinced that the mixing time is improved compared with ULA.
>
> **A2 & A3:** We will remove the claim that CPMLA about weaker assumptions and the mixing time. Instead, we will emphasize that our CPMLA demonstrates a non-bias advantage, while ULA has been shown to exhibit bias in practical settings [1,2].
>
> [1] On the Anatomy of MCMC-Based Maximum Likelihood Learning of Energy-Based Models
> [2] Improved Contrastive Divergence Training of Energy-Based Models

---

### Official Review · Reviewer_wYv9 · 2024-11-04

**Soundness:** 3
**Presentation:** 4
**Contribution:** 3
**Rating:** 8
**Confidence:** 5

**Summary:**

This paper introduces a novel approach for efficiently training and sampling from energy-based models, addressing key limitations such as the challenges in obtaining high-quality samples, which are essential for the training of energy-based models to converge. Additionally, the authors provide valuable theoretical justifications for the convergence of mirror Langevin algorithms within the context of deep neural networks. They demonstrate the effectiveness of their method across several established benchmarks, including CIFAR-10, SVHN, and CelebA, and carefully compare their results with existing approaches.

**Strengths:**

* The proposed method is well-presented, featuring convincing empirical evaluations and solid theoretical justifications.
* The paper is clearly written and effectively organized.
* The proposed algorithm enhances sampling efficiency from energy-based models without significantly increasing the number of parameters in the CP-Flow component.
* The authors include a comprehensive appendix that provides detailed information on the experimental setups and the theoretical proofs.

**Weaknesses:**

* The parameterization of the CP-Flow model in the proposed algorithm appears too constrained, which may complicate the optimization process. Additionally, the authors approximate the Hessian matrix during training using only its diagonal components, and I believe this estimation will become less accurate as the dimensionality increases.

**Questions:**

I am curious about the efficiency of the proposed method in comparison to the approaches presented in [1] and [2], which are relatively easy to implement for both the toy problems and the image generation tasks discussed in the paper. Notably, these methods seem more relevant to the proposed algorithm than some of the literature cited.

[1] Carbone, D., Hua, M., Coste, S., & Vanden-Eijnden, E. (2023). Efficient training of energy-based models using Jarzynski equality. Advances in Neural Information Processing Systems, 36, 52583-52614.

[2] Du, Y., Li, S., Tenenbaum, J., & Mordatch, I. (2020). Improved contrastive divergence training of energy-based models. arXiv preprint arXiv:2012.01316.

---

> ### Author Response · Authors · 2024-11-24
> **Thank you for your insightful comments and positive feedback.**
>
> We greatly appreciate the reviewer’s thoughtful comments and constructive feedback. We are especially grateful for the positive remarks regarding the clear and well-organized presentation of our work, the convincing empirical evaluations, and the solid theoretical justifications.
> Below, we provide detailed responses to the raised questions.
>
> **Q1:** The parameterization of the CP-Flow is too constrained, which may complicate the optimization process.
>
> **A1:** The constraints on the CP-Flow parameters are necessary to ensure that the entire neural network is strongly convex with respect to the input $ x $, thereby satisfying Brenier’s theorem and ensuring optimality toward the target distribution while meeting the conditions for the mirror map [1]. Regarding the optimization process, we follow the approach outlined in [1], which involves conjugate gradient descent and quadratic optimization methods that are computationally efficient and not overly complex. Specifically, we adopt the "input-augmented" version of the Input-Convex Neural Network (ICNN), which enhances the architecture by directly connecting half of the hidden units to the input. This modification introduces a form of skip connection, improving the propagation of gradients during training and alleviating optimization difficulties.
>
> [1] Chin-Wei Huang, Ricky T. Q. Chen, Christos Tsirigotis, and Aaron Courville. Convex potential flows: Universal probability distributions with optimal transport and convex optimization. Learning, 2020.
>
> **Q2:** The approximation of Hessian will become less accurate as the dimensionality increases.
>
> **A2:** The diagonal approximation of the Hessian matrix during training was chosen to balance computational tractability with accuracy. Empirically, we found that this approximation still provides sufficient information for updating the mirror map $ \nabla G $, particularly when combined with stochastic optimization techniques. This is supported by the FID scores and inference speed presented in Table 2 and Figure 2.
>
> **Q3:** What is the efficiency of CPMLA in comparison with the approach presented in [2] and [3]? These methods are more relevant to CPMLA.
>
> **A3:** Empirically, we have shown that our method achieves superior FID scores in real-world image generation tasks compared to both [2] and [3], indicating that CPMLA can outperform the generative quality of models trained using the techniques in those works. Intuitively, our approach operates as a second-order sampling algorithm, which provides both higher sampling efficiency and greater representational power compared to methods relying on pre-determined data augmentations or hierarchical architectures, as used in [2] and [3]. We appreciate the reviewer’s observation that [2] and [3] are closely related to our work. We will add comparisons in the revised manuscript.
>
> [2] Carbone, D., Hua, M., Coste, S., & Vanden-Eijnden, E. (2023). Efficient training of energy-based models using Jarzynski equality. Advances in Neural Information Processing Systems, 36, 52583-52614.
>
> [3] Du, Y., Li, S., Tenenbaum, J., & Mordatch, I. (2020). Improved contrastive divergence training of energy-based models. arXiv preprint arXiv:2012.01316.

---

### Official Review · Reviewer_KVDi · 2024-11-05

**Soundness:** 4
**Presentation:** 3
**Contribution:** 4
**Rating:** 8
**Confidence:** 3

**Summary:**

This paper studies mirror Langevin descent algorithm, where the mirror map is the gradient of a convex function. Novel theoretical results are proven for the convergence of this algorithm under weaker assumptions than previously available in the paper "The Mirror Langevin Algorithm Converges with Vanishing Bias", and in a stronger metric (KL divergence). The mirror map is modelled using an input-convex neural network (Amos et al, 2017), which is formulated in a way to guarantee invertibility of the mirror map. This is called CP-Flow. By jointly optimizing the parameters of this network, and performing sampling, an efficient algorithm is obtained for generative modelling, with dramatically fewer parameters compared to other models (this is due to faster convergence because of the preconditioning of the Langevin dynamics with the mirror map).

**Strengths:**

The paper proposes a very interesting new algorithm for generative modelling based on a combination of input convex neural networks with mirror Langevin descent. There are some new theoretical results showing vanishing bias and bounding convergence rates under weak assumptions. The empirical performance is convincing, especially given the fact that the total number of parameters is much lower than some competing models.

**Weaknesses:**

The FID scores on the example are not yet competitive with the latest score-based generative models such as NCSN++. Nevertheless, there is a lot of potential in this method, and simply modifications might still significantly improve performance.

In terms of presentation, a better explanation of exactly how is the mirror map function G is found in practice would be useful. We have found section 3.2 too short and difficult to fully grasp.

**Questions:**

Could you please give more details on exactly how the mirror map function G is found? Do you alternate between exploring the target distribution, and updating G in an adaptive way, or do you fix G in advance before applying mirror Langevin? You seem to mention cooperative learning on page 6, do you mean that G is updated several times after sampling?

In particular, Section 3.2 should be clarified. Section 4.2 is also not sufficiently clearly explained, especially this cooperative learning aspect.
It would be important to be precise about the implemented algorithm, if there is not sufficient space in the main paper, you could add this discussion in the appendix.

---

> ### Author Response · Authors · 2024-11-24
> **Thank you for your insightful comments and positive feedback.**
>
> We thank the reviewer for their insightful comments and positive feedback on our work. We are gratified that the reviewer recognizes the novelty of both our algorithm and theory, and the compelling empirical performance in terms of both efficiency and generation quality.
> Below, we provide detailed responses to the raised questions.
>
> **Q1:** The FID is not yet competitive with NCSN++.
>
> **A1:** While NCSN++ is a score-based model, CPMLA is an energy-based model (EBM). As a result, a direct comparison between the two may not be quite fair. As shown in Table 2, CPMLA outperforms in terms of FID scores and demonstrates superior efficiency in comparison with previous EBM sampling methods.
>
>
> **Q2:** Is $ G $ updated in an adaptive way, or is it fixed in advance before MLA sampling?
>
> **A2:** As mentioned in Lines 70 and 287, the mirror map $ \nabla G $ in our method is not fixed in advance. Instead, it is optimized jointly with the energy-based model (EBM) during training. Additionally, as noted in Line 389, CPMLA operates with two settings: one where both CP-Flow and the EBM are trained from scratch, and another where CP-Flow is pretrained before training both components together. In both settings, $ G $ is updated adaptively. This adaptive update process, which we refer to as "cooperative learning," is a key innovation of our method. Since $ \nabla G $ acts as a **dynamic mirror map**, it evolves to better capture the geometry of the target distribution throughout the training process.
>
>
> **Q3:** How exactly is the mirror map $ \nabla G $ found in practice?
>
> **A3:** As outlined in Lines 259 and 276 of Algorithm 1, the update of the mirror map $ \nabla G $ alternates between two key steps:
> - **Sampling:** We use the mirror map $ \nabla G $ and the EBM obtained from the previous iteration to perform sampling via the mirror Langevin algorithm (MLA).
> - **Updating $ \nabla G $:** After sampling, the mirror map $ \nabla G $ is updated based on feedback from the target distribution (via the EBM), improving its alignment with the target geometry. This dynamic evolution of $ \nabla G $ ensures that the sampling and optimization reinforce each other throughout the training.
>
> By tightly coupling $ \nabla G $ and the EBM, our method efficiently models complex distributions without relying on a pre-defined or static mirror map.
>
> **Q4:** Sections 3.2 and 4.2 should be explained in more detail, especially the cooperative learning aspect.
>
> **A4:** We sincerely thank the reviewer for their thoughtful suggestions. In line 291 of Section 4.2, we describe the cooperative learning scheme in which we update the parameters $\vartheta$ and $\theta$ using both the original examples $\{x_i\}$ and the synthesized samples $\{x_i^{\text{out}}\}$ based on Equations 6 and 3. We will provide more detailed information, including the practical steps for optimization, the frequency of $ G $ updates relative to the sampling steps, and the role of the feedback loop between $ G $ and the EBM in enhancing sampling performance. Additionally, we will include a diagram and an algorithm to illustrate the cooperative learning process.

---

> > ### Comment · Reviewer_KVDi · 2024-11-24
> > **Thanks for the response**
> >
> > Thanks for the response. Including these additional details is welcome. I keep my mark.

---

### Author Response · Authors · 2024-12-04
**Common response to all reviewers and ACs**

We sincerely thank the reviewers for their positive feedback on the novelty and effectiveness of the proposed CPMLA and their constructive suggestions on the theoretical and experimental improvement. We summarize the reviewers' comments and our rebuttals as follows.

### Summary of Reviewer Feedback:

#### Strengths:
- **Novelty**: Reviewers acknowledged the innovative approach of combining mirror Langevin dynamics with convex potential flow, as well as the novel theoretical results that prove exponential convergence under relaxed conditions.
- **Empirical Results**: Our empirical evaluation, including comparisons with established benchmarks, was positively noted, especially regarding CPMLA’s ability to achieve high sampling efficiency and quality with fewer parameters compared to other methods.
- **Presentation**: The clarity and organization of the paper were appreciated, with particular praise for the clear motivation and sound theoretical foundations.

#### Suggested Revisions:
We have carefully considered the points raised by the reviewers and present the key revisions that will be incorporated in the revised manuscript:

**Clarity on Mirror Map Calculation**:
  We will enhance Section 3.2 by adding a detailed explanation of how the mirror map $\nabla G$ is computed, including how the optimization is performed. We will elaborate on how the mirror map $\nabla G$ is updated adaptively during the training process, as mentioned in Lines 70 and 287, with references to the cooperative learning mechanism described in Line 389.

**FID Scores and Comparisons**:
  We will include further experiments comparing CPMLA with more relevant models and present results with extended iterations to highlight CPMLA's competitive advantages over wall clock time.

  **Table 1:** A comparison of wall clock time for inferring 1,000 images of similar quality on the CIFAR-10 dataset
|                    | CPMLAprt (T = 20) | CoopFlow (T = 30) |
|:------------------:|:-----------------:|:-----------------:|
| Time (s/1k images) | 16.14             | 16.84             |

**Table 2:** A comparison of FID scores under different iteration counts
|                   | 10    | 20    | 24    | 27    | 30    |
|:-----------------:|:-----:|:-----:|:-----:|:-----:|:-----:|
| CoopFlow (T = 30) | 57.26 | 30.74 | 25.03 | 22.45 | 21.16 |
| CPMLA (T = 20)    | 56.40 | 26.92 | 24.76 | 23.47 | 22.35 |

  We will also include a comparison with methods of similar parameter counts to further demonstrate the efficiency of our approach.

**Table 3:** A comparison of FID scores and the number of parameters across different EBMs and flow-based frameworks
| Methods       | Number of Parameters | FID   |
|:-------------:|:--------------------:|:-----:|
| NT-EBM        | 23.8M                | 78.12 |
| GLOW          | 44.2M                | 45.99 |
| EBM_FCE       | 44.9M                | 37.30 |
| CoopFlow      | 45.9M                | 21.16 |
| Flow++ only   | 28.8M                | 92.10 |
| VAEBM         | 135.1M               | 12.16 |
| **CPMLA**     | **17.39M**           | **22.35** |

**Theoretical Extentions**:
  We will revise Section 5 to provide a more nuanced explanation regarding the approximation error and its impact on inference efficiency. We will clarify that our advantages on relaxed log-concavity assumption and mixing time, and update Theorem 5.5 to prove that the total variation distance $d_{TV}(\rho_t, p_{\text{data}}) < \delta$.

**Hessian Approximation and Optimization**:
  We will add a discussion detailing the advantages and limitations of the diagonal Hessian approximation used in our method, including its benefits for computational efficiency, particularly in high-dimensional settings where full Hessian calculations are not feasible. This will be addressed near Algorithm 1.

**Typo Clarification**:
  We will ensure that all mathematical formulations in Algorithm 1 are accurate and clearly explained. We will correct the identified typo in Line 284, where the gradient is evaluated in the dual space $y - \eta \nabla f_{\theta}(\nabla G^* y) + \sqrt{2\eta}\xi$, and revise notations in Equation 8 $\nabla^2 G(\nabla G^* y_t)$. These clarifications will help ensure that readers can easily understand the algorithm's implementation.

---

### Meta-Review · Area_Chair_Sqtp · 2024-12-17

**Metareview:**

The authors introduced a new method for sampling from energy based models, where the algorithm trains a dynamic mirror map, before using a mirror based Langevin algorithm to sample from the energy. The reviews here have very high variance, and made the decision somewhat unclear. On the positive side, the algorithm appears to be novel, the authors performed extensive empirical evaluation, and the performance is good albeit not quite state of the art. On the negative side, an extended discussion with a reviewer led to a lot of confusion on both the implementation and the theoretical aspects of this paper.

Most importantly, this discussion did not resolve all of the original concerns. And without access to the significant revisions to the manuscript on both the theoretical front, and clarity into the implementation of the algorithm, many of these concerns cannot be resolved at this point. Unlike the review process for journals, the reviewers at ICLR cannot request a major revision to further continue the discussion and improve the manuscript. Therefore, I believe the paper in its current form is not ready to be accepted.

Given that I also personally find many components of this paper confusing to read myself, I would encourage the authors try to reformulate the story and technical explanations of this paper before resubmitting. There is a potentially very interesting algorithm with theoretical results here, and an improved manuscript should be a welcomed addition to the energy based models literature.

**Additional Comments On Reviewer Discussion:**

The most significant discussion is with reviewer 67JH, where I believe a lot of confusion was uncovered throughout the process on both the implementation and the theory of the algorithm. For example, the proof of Theorem 5.5 was revised, and a proof sketch is provided. This makes a theoretical verification at the conference level difficult, as we do not have the luxury of major revisions for such changes.

---

### Decision · Program_Chairs · 2025-01-22

Reject